# COMBINATORIAL DUELING BANDITS

## ABSTRACT

We introduce the *Contextual Combinatorial Dueling Bandits (CDB)* problem, a novel framework for modeling complex online decision-making under relative and binary feedback. In each round, the learner observes contextual information for a set of arms and selects two subsets of $k$ arms, termed *super arms*. The feedback consists of pairwise binary preferences between the arms in the two chosen super arms. For example, in recommendation systems, a user might be shown two competing sets of items and provide preference feedback for each pair of items. We propose two algorithms to address this problem: *LinCDB* for linear score functions and *NCDB* for nonlinear cases. Both algorithms leverage the Hungarian algorithm for efficient selection of the second super arm. We theoretically demonstrate that LinCDB achieves a regret bound of $\widetilde{O}\left(\frac{d}{\kappa_\mu}\sqrt{Tk}\right)$, while NCDB achieves $\widetilde{O}\left(\left(\frac{1}{\kappa_\mu}\sqrt{\widetilde{d}} + B\sqrt{\frac{\lambda}{\kappa_\nu}}\right)\sqrt{Tk\widetilde{d}}\right)$. Here, $d$ represents the dimension of the context for each arm, $k$ is the size of the super arm, and $\widetilde{d}$ denotes the effective dimension. To our knowledge, this is the first work to study combinatorial bandits with preference feedback.

## 1 INTRODUCTION

*Combinatorial bandits* offer a powerful framework for sequential decision-making in domains like recommendation systems, ad placement, and medical diagnosis (Saha & Gopalan, 2019; Nika et al., 2020; Hwang et al., 2023). In the standard setting, a learning agent selects a "super arm" (a subset of items) and observes semi-bandit feedback, meaning it receives a numerical reward for each individual item chosen (Qin et al., 2014; Chen et al., 2018). However, this assumption of observing explicit numerical scores is often impractical. In many real-world applications, especially those involving human interaction such as recommender systems or Reinforcement Learning from Human Feedback (RLHF) for LLMs, feedback is more readily available as *relative preferences between items*.

To account preference feedback, the *dueling bandit* framework models feedback as pairwise preferences, and its contextual variant has proven highly effective in capturing human choices (Saha, 2021; Saha & Krishnamurthy, 2022; Bengs et al., 2022; Li et al., 2024; Verma et al., 2025). Motivated by scenarios that require both combinatorial choices and preference-based feedback, we introduce and study a more general and realistic problem: the *combinatorial dueling bandit (CDB)*. In our setting, an agent is presented with $N$ arms in each round, each described by a $d$-dimensional context vector. The agent's task is to select two super arms, each containing $k$ base arms. It then forms $k$ pairs between the arms of the two super arms and observes binary preference feedback for each pair. The agent receives an overall numerical reward for each of the two chosen super arms, and the ultimate goal is to design a policy that maximizes the cumulative reward over time.

We model this process by assuming a latent scoring function that assigns a utility score to each arm. Both the binary preference outcomes and the super arm rewards are functions of these underlying scores. Specifically, we adopt the well-established Bradley–Terry–Luce (BTL) model (Luce, 2005; Saha, 2021; Bengs et al., 2022), where the probability of one arm being preferred over another is determined by their exponentiated scores (Sec. 2). This CDB setting naturally models numerous applications. For instance, in recommendation systems, a user might be shown two competing slates of items and provide preference feedback to pairs of items. In LLM training, responses from multiple models can be paired and ranked by human evaluators. Similarly, in online advertising, the effectiveness of two different ad campaigns (i.e., sets of ads) can be compared.

This paper makes several key contributions. We first formally define the contextual combinatorial dueling bandit problem. We then propose two novel algorithms to solve it:

1. **LinCDB** (**Lin**ear **C**ombinatorial **D**ueling **B**andits) for settings where the latent score function is linear, which works by minimizing a cross-entropy loss based on the BTL model.

2. **NCDB** (**N**eural **C**ombinatorial **D**ueling **B**andits) for general nonlinear score functions, which leverages a neural network to approximate the score function within an Upper Confidence Bound (UCB) framework (Auer et al., 2002; Chu et al., 2011; Abbasi-Yadkori et al., 2011; Zhou et al., 2020; Chowdhury & Gopalan, 2017).

A significant technical challenge in our setting is the selection of the second super arm. While the first super arm can be chosen greedily by picking the top-$k$ arms based on estimated scores, an optimal choice for the second super arm—to balance exploration and exploitation—would require searching through $O(N^k)$ combinations. We circumvent this computational hurdle by framing the selection of the second super arm as a *bipartite matching problem*, which can be solved efficiently in polynomial time using the Hungarian algorithm (see Appendix C).

We provide rigorous theoretical analyses for both algorithms, establishing their regret bounds. We prove that LinCDB achieves a regret of $\widetilde{O}(\frac{d}{\kappa_\mu}\sqrt{Tk})$ and NCDB achieves a regret of $\widetilde{O}\left(\left(\frac{1}{\kappa_\mu}\widetilde{d} + B\sqrt{\frac{\lambda}{\kappa_\nu}}\right)\sqrt{Tk\widetilde{d}}\right)$, where $d$ is the context dimension, $k$ is the super arm size, and $\widetilde{d}$ is the effective dimension. The practical performance of our algorithms is validated through a series of synthetic experiments. To the best of our knowledge, *this is the systematic study of contextual combinatorial dueling bandits*, and we consider both linear and general nonlinear score functions. We further extend our discussion of the reward function under the Lipschitz continuity assumption in Appendix F.

## 2 PROBLEM SETTING

**Contextual combinatorial dueling bandits.** We study the contextual combinatorial bandit problem with dueling feedback, where the learner selects two subsets of arms and receives pairwise comparison feedback between each matched pair of arms from the two subsets. Our setting differs from the standard contextual combinatorial bandits, where the learner selects a single set of arms and receives individual reward feedback for each selected arm.

At each round $t$, the learner observes a set of context vectors $\mathcal{X}_t = \{\mathbf{x}_{t,1}, \mathbf{x}_{t,2}, \ldots, \mathbf{x}_{t,N}\} \subset \mathcal{X} \subset \mathbb{R}^d$ corresponding to $N$ arms where $\mathcal{X}$ denote the global context space. The learner then selects two ordered sets of arms $S_t^1, S_t^2 \subset [N]$, $|S_t^1| = |S_t^2| = k$ referred to as *super arms*. We denote $S_t^1 = \{s_{t,1}^1, s_{t,2}^1, \ldots, s_{t,k}^1\}, S_t^2 = \{s_{t,1}^2, s_{t,2}^2, \ldots, s_{t,k}^2\}$ and $\mathcal{X}_t(s_{t,i}^p) = \mathbf{x}_{t,s_{t,i}^p}$ denote the context of $s_{t,i}^p - th$ arm from $\mathcal{X}_t$. For simplicity, we use the notation $\mathbf{x}_{t,i}^p$ instead of $\mathbf{x}_{t,s_{t,i}^p}$, i.e. $\mathbf{x}_{t,i}^p = \mathbf{x}_{t,s_{t,i}^p}$. Let $\mathcal{S} = \{S \subset [N] \big| |S| = k\}$ denote the set of all candidate super arms of size $k$ and $\mathcal{S} \subset 2^{[N]}$. Note that $S_t^1$ and $S_t^2$ may share overlapping arms, i.e., $S_t^1 \cap S_t^2 \neq \emptyset$ is allowed.

After choosing $S_t^1$ and $S_t^2$, the learner receives a set of stochastic pairwise preference feedback: $\{y_{t,1}, y_{t,2}, \ldots, y_{t,k}\}$, where each $y_{t,i} \in \{0,1\}$ represents the outcome of a noisy comparison between $\mathbf{x}_{t,i}^1$ and $\mathbf{x}_{t,i}^2$. Specifically, $y_{t,i} = 1$ indicates that $\mathbf{x}_{t,i}^1$ is preferred over $\mathbf{x}_{t,i}^2$, and $y_{t,i} = 0$ otherwise. The feedback depends on an underlying *latent score function* $r^* : \mathcal{X} \to \mathbb{R}$, which is unknown to the learner. Let $\mathbf{r}_t^* = [r^*(\mathbf{x}_{t,1}), \ldots, r^*(\mathbf{x}_{t,N})] \in \mathbb{R}^N$ be the latent scores of all arms at round $t$. Then the score for arm $i$ is defined as: $r_{t,i}^* = r^*(\mathbf{x}_{t,i})$. Given the unknown score vector $\mathbf{r}_t^*$ and the selected super arms $S_t^1$ and $S_t^2$, the learner obtains rewards $f(S_t^1, \mathbf{r}_t^*)$ and $f(S_t^2, \mathbf{r}_t^*)$, which we discuss next.

**Reward function.** We consider a deterministic reward function $f(S, \mathbf{r})$ that evaluates the quality of a super arm $S \subset [N]$ based on a score vector $\mathbf{r} \in \mathbb{R}^N$. Specifically, we assume the reward of a super arm is the sum of the scores of its constituent arms. That is, the reward function is defined as $f(S, \mathbf{r}) = \sum_{i \in S} r_i$ which is consistent with previous work (Wen et al., 2015; Kveton et al., 2015; Louëdec et al., 2015). This additive reward formulation captures a wide range of applications where

the overall utility is naturally decomposable across selected arms. We further extend our discussion of the reward function under the Lipschitz continuity assumption in Appendix F.

**Stochastic preference model.** We model the pairwise preference feedback as a Bernoulli random variable governed by the Bradley–Terry–Luce (BTL) model (Hunter, 2004; Luce, 2005), a widely adopted framework in the study of dueling bandit problems (Saha, 2021; Bengs et al., 2022; Li et al., 2024). Under this model, the probability that the arm $\mathbf{x}_{t,i}^1$ in the first selected super arm is preferred over $\mathbf{x}_{t,i}^2$ in the second selected super arm, conditioned on the context $\mathcal{X}_t$ and the underlying latent score function $r^*$, is given by

$$\mathbb{P}\{x_{t,i}^1 \succ x_{t,i}^2\} = \mathbb{P}\{y_{t,i} = 1 \mid x_{t,i}^1, x_{t,i}^2\} = \frac{\exp(r^*(x_{t,i}^1))}{\exp(r^*(x_{t,i}^1)) + \exp(r^*(x_{t,i}^2))} = \mu(r^*(x_{t,i}^1) - r^*(x_{t,i}^2)),$$

where $\mu(\cdot)$ denotes the logistic link function. This formulation captures the relative preference between two arms based on their latent utilities, and aligns with established stochastic choice theory.

Here, $x_{t,i}^1 \succ x_{t,i}^2$ denotes that arm $x_{t,i}^1$ is preferred over arm $x_{t,i}^2$, $\mu(x) = 1/(1 + e^{-x})$ represents the sigmoid (logistic) function, and $r^*(x_{t,i})$ denotes the latent utility associated with the $i$-th selected arm. While our analysis primarily adopts the Bradley–Terry–Luce (BTL) model, the results are applicable to a broader class of stochastic preference models (Bengs et al., 2022).

To ensure the generality of our theoretical guarantees across different preference models, we impose a set of regularity conditions on the function $\mu(\cdot)$, also referred to as the *link function* (Li et al., 2017; Bengs et al., 2022):

**Assumption 1.** *We assume the following:*

- $\kappa_\mu = \inf_{\mathbf{x},\mathbf{x}' \in \mathcal{X}} \dot{\mu}(r^*(\mathbf{x}) - r^*(\mathbf{x}')) > 0$ *for all pairs of context-arm.*

- *The link function $\mu : \mathbb{R} \to [0,1]$ is continuously differentiable and Lipschitz with constant $L_\mu$.*

- $\|\mathbf{x} - \mathbf{x}'\| \le D, \forall \mathbf{x}, \mathbf{x}' \in \mathcal{X}.$

**Performance measure.** After selecting two super arms, denoted by $S_t^1$ and $S_t^2$ in round $t$, the learner incurs an *instantaneous regret*. We define the *optimal super arm* at round $t$ as $S_t^* = \arg\max_{S \in \mathcal{S}} f(S, \mathbf{r}_t^*)$, where $\mathbf{r}_t^* = [r^*(\mathbf{x}_{t,i})]_{i=1,\dots N}$. Similar to standard dueling bandit settings, there are two commonly used notions of instantaneous regret in the *combinatorial dueling bandit* setting (Saha, 2021; Bengs et al., 2022; Li et al., 2024; Verma et al., 2025): the *average instantaneous regret*: $reg_t^a = f(S_t^*, \mathbf{r}_t^*) - \frac{1}{2}\left(f(S_t^1, \mathbf{r}_t^*) + f(S_t^2, \mathbf{r}_t^*)\right)$, and the *weak instantaneous regret*: $reg_t^w = f(S_t^*, \mathbf{r}_t^*) - \max\left\{f(S_t^1, \mathbf{r}_t^*), f(S_t^2, \mathbf{r}_t^*)\right\}$. Accordingly, the *cumulative regret* over $T$ rounds is defined as $Reg_T^\tau = \sum_{t=1}^T reg_t^\tau$, where $\tau \in \{a, w\}$. Note that $Reg_T^w \le Reg_T^a$, so an upper bound on $Reg_T^a$ also serves as an upper bound on $Reg_T^w$. Therefore, in the subsequent analysis (Secs. 3.2 and 4.3), we will focus on deriving an upper bound on $Reg_T^a$, which we will denote as $Reg_T$ for simplicity.

## 3 LINEAR COMBINATORIAL DUELING BANDITS

In this section, we assume the unknown score function $r^* : \mathcal{X} \to \mathbb{R}$ to be linear. Formally, $r^*(\mathbf{x}) = \theta^\top \mathbf{x}$ where $\theta$ are unknown parameters. And denote our estimation of the unknown function in each iteration $t$ by $r_t(\mathbf{x}) = \theta_t^\top \mathbf{x}$. Based on this linear model, we propose an algorithm for the linear contextual combinatorial dueling bandit problem.

### 3.1 THE LINCDB ALGORITHM

We present our first algorithm: Linear Combinatorial Dueling Bandits (LinCDB) in Algorithm 1.

---

**Algorithm 1** Linear Combinatorial Dueling Bandits (LinCDB)

---

1: Set $V_0 \triangleq \frac{\lambda}{\kappa_\mu} \mathbf{I}$, $\beta_t \triangleq \sqrt{2 \log(1/\delta) + d \log\left(1 + tkD^2 \kappa_\mu/(d\lambda)\right)}$.
2: **for** $t = 1, \ldots, T$ **do**
3:     Find $\theta_t = \arg\min_{\theta'} \mathcal{L}_t(\theta')$ equation 1
4:     Choose the first super arm $S_t^1 = \arg\max_{S \in \mathcal{S}} f(S, \mathbf{r}_t)$
5:     **for** $i = 1, \ldots, k$ **do**
6:        **for** $j = 1, \ldots, N$ **do**
7:           $\text{value}(\mathbf{x}_{t,i}^1, \mathbf{x}_{t,j}) = r_t(\mathbf{x}_{t,j}) + \frac{\beta_t}{\kappa_\mu} \left\| \mathbf{x}_{t,i}^1 - \mathbf{x}_{t,j} \right\|_{V_{t-1}^{-1}}$.
8:        **end for**
9:     **end for**
10:     Choose the second super arm $S_t^2 = \arg\max_{S \in \mathcal{S}}[f(S, \mathbf{r}_t) + \frac{\beta_t}{\kappa_\mu} \sigma_{\mathcal{X}_t}(S, S_t^1)]$ via $\text{value}(\mathbf{x}_{t,i}^1, \mathbf{x}_{t,j})$ with Hungarian Algorithm in Algorithm 3.
11:     Observe the preference feedback: $\{y_{t,i} = \mathbb{K}(x_{t,i}^1 \succ x_{t,i}^2)\}_{i=1,\ldots k}$, and update history
12:     Update $V_t \leftarrow V_{t-1} + \sum_{i=1}^k \widetilde{\mathbf{x}}_{t,i} \widetilde{\mathbf{x}}_{t,i}^\top$
13: **end for**

---

At each iteration $t$, we first estimate the parameter $\theta_t$ by minimizing the following loss function with historical observations $\{(\mathbf{x}_{s,i}^1, \mathbf{x}_{s,i}^2, y_{s,i})\}_{s=1,\ldots t-1, i=1,\ldots k}$:

$$
\mathcal{L}_t(\theta') = -\sum_{s=1}^{t-1} \sum_{i=1}^{k} \left[ y_{s,i} \log \mu \left( {\theta'}^\top \left[ \mathbf{x}_{s,i}^1 - \mathbf{x}_{s,i}^2 \right] \right) \right.
$$
$$
\left. + (1 - y_{s,i}) \log \mu \left( {\theta'}^\top \left[ \mathbf{x}_{s,i}^2 - \mathbf{x}_{s,i}^1 \right] \right) \right] + \frac{1}{2} \lambda \left\| \theta' \right\|_2^2 . \tag{1}
$$

Formally, $\theta_t = \arg\min_{\theta'} \mathcal{L}_t(\theta')$ corresponds to the maximum likelihood estimate of the unknown parameter $\theta$ based on the observed history. When the loss function equation 1 is minimized exactly (i.e., the gradient is zero), the following optimality condition holds:

$$
\sum_{s=1}^{t-1} \sum_{i=1}^{k} \left( \mu \left( \theta_t^\top \left[ \mathbf{x}_{s,i}^1 - \mathbf{x}_{s,i}^2 \right] \right) - y_{s,i} \right) \left[ \mathbf{x}_{s,i}^1 - \mathbf{x}_{s,i}^2 \right] + \lambda \theta_t = 0, \tag{2}
$$

which is a crucial step in our analysis.

At iteration $t$, upon receiving the context $\mathcal{X}_t$, the learner selects two super arms $S_t^1, S_t^2 \subset [N]$ of size $k$ and observes preference feedback $\{y_{t,1}, \ldots, y_{t,k}\}$. The context of super arm is denoted by $\mathcal{X}_t(S_t^1) = \{\mathbf{x}_{t,1}^1, \ldots \mathbf{x}_{t,k}^1\}, \mathcal{X}_t(S_t^2) = \{\mathbf{x}_{t,1}^2, \ldots \mathbf{x}_{t,k}^2\}$. For each pair of arms $\mathbf{x}_{t,i}^1$ and $\mathbf{x}_{t,i}^2$, we collect preference feedback $y_{t,i} = \mathbb{K}(\mathbf{x}_{t,i}^1 \succ \mathbf{x}_{t,i}^2)$, which is equal to 1 if $\mathbf{x}_{t,i}^1$ is preferred over $\mathbf{x}_{t,i}^2$ and 0 otherwise.

The first super arm $S_t^1$ is chosen greedily by maximizing the reward function as follows:

$$
S_t^1 = \arg\max_{S \in \mathcal{S}} f(S, \mathbf{r}_t), \tag{3}
$$

in which $\mathbf{r}_t$ is the estimated score of all the arms using function $r_t(\mathbf{x}) = \theta_t^\top \mathbf{x}$ in iteration $t$. After that, the second arm $S_t^2$ is selected using the UCB algorithm:

$$
S_t^2 = \arg\max_{S \in \mathcal{S}}[f(S, \mathbf{r}_t) + \frac{\beta_t}{\kappa_\mu} \sigma_{\mathcal{X}_t}(S, S_t^1)], \tag{4}
$$

in which we denote $\mathcal{X}_t(S) = \{\mathbf{x}_1, \ldots \mathbf{x}_k\}$ and

$$
\sigma_{\mathcal{X}_t}(S, S_t^1) \triangleq \sum_{i=1}^{k} \left\| \mathbf{x}_i - \mathbf{x}_{t,i}^1 \right\|_{V_{t-1}^{-1}} . \tag{5}
$$

Here $V_t = \sum_{s=1}^t \sum_{i=1}^k \widetilde{\mathbf{x}}_{s,i} \widetilde{\mathbf{x}}_{s,i}^\top + \frac{\lambda}{\kappa_\mu} \mathbf{I}$ and $\widetilde{\mathbf{x}}_{s,i} = \mathbf{x}_{s,i}^1 - \mathbf{x}_{s,i}^2$ which corresponds to the context of arm $s_{t,i}^1 \in S_t^1$ and $s_{t,i}^2 \in S_t^2$. Intuitively, the first super arm $S_t^1$ is chosen greedily by choosing

the top $k$ arms with the highest estimated scores equation 3. After selecting $S_t^1$, the uncertainty measure $\sigma_{\mathcal{X}_t}(S, S_t^1)$ tends to be larger for super arms $S$ that differ more from $S_t^1$ given the historical feedback. Therefore, the selection of the second arm (line 5 of Algo. 1) is able to balance *exploration and exploitation*.

However, choosing the second super arm $S_t^2$ is more challenging, as a naive search over all possible super arms $S \in \mathcal{S}$ may require exponential time. Fortunately, in our setting, the combined objective can be decomposed as

$$
f(S_t^2, \mathbf{r}_t) + \frac{\beta_t}{\kappa_\mu} \sigma_{\mathcal{X}_t}(S_t^2, S_t^1) = \sum_{i=1}^{k} \left( r_t(\mathbf{x}_{t,i}^2) + \frac{\beta_t}{\kappa_\mu} \left\| \mathbf{x}_{t,i}^1 - \mathbf{x}_{t,i}^2 \right\|_{V_{t-1}^{-1}} \right)
$$

$$
= \sum_{i=1}^{k} \text{value}(\mathbf{x}_{t,i}^1, \mathbf{x}_{t,i}^2).
$$

We decompose the complex maximization objective into independent terms $\text{value}(\mathbf{x}_{t,i}^1, \mathbf{x}_{t,i}^2)$, reducing the problem of selecting the best super arm to choosing $\mathbf{x}_{t,i}^2$ that maximizes $\sum_{i=1}^{k} \text{value}(\mathbf{x}_{t,i}^1, \mathbf{x}_{t,i}^2)$. Since $S_t^1$ is fixed and each $\mathbf{x}_{t,i}^1$ is known, $\text{value}(\mathbf{x}_{t,i}^1, \mathbf{x}_{t,i}^2)$ depends only on $\mathbf{x}_{t,i}^2$, and can therefore be computed efficiently. This allows us to independently select each $\mathbf{x}_{t,i}^2$ to construct the optimal second super arm $S_t^2$. Further details are provided in Appendix C.

### 3.2 Regret Analysis

**Lemma 1.** *In each iteration $t = 1, \ldots, T$, for $\mathbf{x}_1, \mathbf{x}_2 \in \mathcal{X}_t$ with probability of at least $1 - \delta$, we have that*

$$
| (r^*(\mathbf{x}_1) - r^*(\mathbf{x}_2)) - (r_t(\mathbf{x}_1) - r_t(\mathbf{x}_2)) | \leq \frac{\beta_t}{\kappa_\mu} \left\| \mathbf{x}_1 - \mathbf{x}_2 \right\|_{V_{t-1}^{-1}}
$$

**Theorem 1** (LinCDB). *Let $\lambda \leq \frac{\kappa_\mu}{L^2}$ and $\beta_t \triangleq \sqrt{2 \log(1/\delta) + d \log\left(1 + tkD^2\kappa_\mu/(d\lambda)\right)}$, then with probability of at least $1 - \delta$, we have that*

$$
Reg_T \leq \frac{3}{\kappa_\mu} \sqrt{2 \log(1/\delta) + d \log\left(1 + TkD^2\kappa_\mu/(d\lambda)\right)} \sqrt{2Tkd \log(1 + TkD^2\kappa_\mu/(d\lambda))}
$$

*Ignoring all log factors, we have that: $Reg_T = \widetilde{O}(\frac{d}{\kappa_\mu}\sqrt{Tk})$.*

Theorem 1 establishes that the cumulative regret of LinCDB is bounded by $Reg_T = \widetilde{O}\left(\frac{d}{\kappa_\mu}\sqrt{Tk}\right)$, which is sublinear in $T$. The dependence on the parameter $k$ reflects the cost of performing $k$ comparisons per round, as is expected in the combinatorial setting. The dependence on $1/\kappa_\mu$ is consistent with prior work on dueling bandits (Bengs et al., 2022) and reflects the cost associated with having less informative dueling feedback compared to numerical feedback. A detailed proof of Theorem 1 is provided in the Appendix. We also provide an analysis of the regret under the Lipschitz continuity assumption in Appendix F.

## 4 Neural Combinatorial Dueling Bandits

In this section, we drop the assumption that the unknown score function $r^* : \mathcal{X} \to \mathbb{R}$ is linear which is required by LinCDB. Instead, we consider a more general setting where $r^*$ can be a non-linear function, and we adopt a neural network model to approximate it.

### 4.1 Neural Network

We use a fully connected neural network to estimate the non-linear score function $r^*$ with depth $L \geq 2$ and the width $m$ (Zhou et al., 2020; Zhang et al., 2021). And let $h(\mathbf{x}; \theta)$ represent the output of the neural network with context input $\mathbf{x}$ and parameter $\theta$:

$$
h(\mathbf{x}; \theta) = \mathbf{W}_L \operatorname{ReLU}\left(\mathbf{W}_{L-1} \operatorname{ReLU}\left(\cdots \operatorname{ReLU}(\mathbf{W}_1 \mathbf{x})\right)\right)
$$

where $\text{ReLU}(x) = \max\{x, 0\}$. The weight matrices are defined as follows: $\mathbf{W}_1 \in \mathbb{R}^{m \times d}$ for the input layer, $\mathbf{W}_\ell \in \mathbb{R}^{m \times m}$ for hidden layers with $2 \leq \ell < L$, and $\mathbf{W}_L \in \mathbb{R}^{1 \times m}$ for the output layer.

We denote the collection of all network parameters by

$$\theta := \left[ \text{vec}(\mathbf{W}_1)^\top, \ldots, \text{vec}(\mathbf{W}_L)^\top \right]^\top \in \mathbb{R}^p,$$

where $\text{vec}(\cdot)$ denotes the vectorization operator, and the total number of parameters is $p = dm + m^2(L-2) + m$.

In every iteration $t$, we solve for $\theta_t$ by minimizing the following loss function:

$$\mathcal{L}_t(\theta) = -\sum_{s=1}^{t-1} \sum_{i=1}^{k} (y_{s,i} \log \mu \left[ h(\mathbf{x}_{s,i}^1; \theta) - h(\mathbf{x}_{s,i}^2; \theta) \right]$$

$$+ (1 - y_{s,i}) \log \mu \left[ h(x_{s,i}^1; \theta) - h(\mathbf{x}_{s,i}^2; \theta) \right]) + \frac{1}{2} \lambda \left\| \theta - \theta_0 \right\|_2^2, \tag{6}$$

in which $\theta_0$ denotes the initial parameters of the neural network, which are initialized following the approach used in prior work (Zhou et al., 2020; Zhang et al., 2021). We denote by $\theta_t$ the estimate of the parameter $\theta$ at iteration $t$.

## 4.2 THE NCDB ALGORITHM

Here, we introduce our second algorithm, NCDB, which is designed for scenarios where the score function is nonlinear. Adopting a similar UCB-based super arm selection strategy as our LinCDB algorithm (Section 3), NCDB differs from LinCDB by employing a neural network to estimate the score function $r^*$.

---

**Algorithm 2** Neural Combinatorial Dueling Bandits (NCDB)

---

1: Set $V_0 \triangleq \frac{\lambda}{\kappa_\mu} \mathbf{I}$, $\beta_T \triangleq \frac{1}{\kappa_\mu} \sqrt{\widetilde{d} + 2\log(1/\delta)}$ ($\widetilde{d}$ is defined in Definition 1), $\nu_T \triangleq \left( \beta_T + B\sqrt{\frac{\lambda}{\kappa_\mu}} + 1 \right) \frac{\kappa_\mu}{\lambda}$.

2: **for** $t = 1, \ldots, T$ **do**

3:      Train NN using history $\{(\mathbf{x}_{s,i}^1, \mathbf{x}_{s,i}^2, y_{s,i})\}_{s=1, i=1}^{t-1, k}$ by minimizing loss function equation 6

4:      Receive the contexts $\mathcal{X}_t$

5:      Compute $r_t(x) = h(x; \theta_t)$

6:      Choose the first arm set $S_t^1 = \arg\max_{S \in \mathcal{S}} f(S, \mathbf{r}_t)$

7:      **for** $i = 1, \ldots, k$ **do**

8:         **for** $j = 1, \ldots, N$ **do**

9:             $\text{value}(\mathbf{x}_{t,i}^1, \mathbf{x}_{t,j}) = r_t(\mathbf{x}_{t,j}) + \frac{\beta_t}{\kappa_\mu} \left\| \mathbf{x}_{t,i}^1 - \mathbf{x}_{t,j} \right\|_{V_{t-1}^{-1}}$.

10:         **end for**

11:      **end for**

12:      Choose the second super arm $S_t^2 = \arg\max_{S \in \mathcal{S}} f(S, \mathbf{r}_t) + \nu_T \sigma_{t-1}(S, S_t^1)$ via $\text{value}(\mathbf{x}_{t,i}^1, \mathbf{x}_{t,j})$ with Hungarian Algorithm in Algorithm 3.

13:      Observe the preference feedback: $\{y_{t,i} = \Bbbk(x_{t,i}^1 \succ x_{t,i}^2)\}_{i=1,\ldots k}$, and update history

14: **end for**

---

We use $g(\mathbf{x}; \theta)$ to denote the gradient of the neural network with respect to parameters $\theta$ at input $\mathbf{x}$. Furthermore, we denote $\phi(\mathbf{x}) = g(\mathbf{x}; \theta_0)$, and define $\widetilde{\phi}(\mathbf{x}_{\tau,i}) = \phi(\mathbf{x}_{\tau,i}^1) - \phi(\mathbf{x}_{\tau,i}^2) = g(\mathbf{x}_{\tau,i}^1; \theta_0) - g(\mathbf{x}_{\tau,i}^2; \theta_0)$. With these definitions, $g(\mathbf{x}; \theta_0)/\sqrt{m}$ represents the random feature approximation of the context-arm feature vector $\mathbf{x}$ with respect to the neural tangent kernel (NTK) (Zhou et al., 2020). We further define the feature covariance matrix $V_t \in \mathbb{R}^{d \times d}$ as

$$V_t = \sum_{\tau=1}^{t} \sum_{i=1}^{k} \frac{1}{m} \widetilde{\phi}(\mathbf{x}_{\tau,i}) \widetilde{\phi}(\mathbf{x}_{\tau,i})^\top + \frac{\lambda}{\kappa_\mu} \mathbf{I}, \tag{7}$$

which will be used in NCDB algorithm and in the regret bound analysis.

In addition, we define the following uncertainty term for a pair of super arms $(S_t^1, S_t^2)$:

$$\sigma_t(S_t^1, S_t^2) \triangleq \frac{\lambda}{\kappa_\mu} \sum_{i=1}^k \left\| \frac{1}{\sqrt{m}} \left( \phi(\mathbf{x}_{t,i}^1) - \phi(\mathbf{x}_{t,i}^2) \right) \right\|_{V_{t-1}^{-1}}, \tag{8}$$

which is used to select the second super arm $S_t^2$ (see line 7 of Algo. 2).

With the definitions above, we present the proposed UCB-based Neural Combinatorial Dueling Bandits (NCDB) algorithm in Algo. 2. Similar to LinCDB, in NCDB (Algo. 2), we select the first super arm $S_t^1$ via greedy selection, and choose the second super arm $S_t^2$ by balancing exploration and exploitation.

### 4.3 REGRET ANALYSIS

Let $N$ denote the number of arms in each round, and let $\mathbf{H} = \frac{1}{2} \left( \mathbf{H}^{(L)} + \mathbf{\Sigma}^{(L)} \right)$ be the *neural tangent kernel (NTK)* matrix constructed over the context set $\{\mathbf{x}_k\}_{k=1}^{TN}$ Zhou et al. (2020). The matrix $\mathbf{H}$ is defined recursively from the input layer to the output layer of the neural network, following the construction in (Zhou et al., 2020; Zhang et al., 2021); a detailed definition is provided in Definition 2 in the appendix. We denote the $j$-th element of the vector $\mathbf{x}$ by $x_j$. We now introduce the assumptions required in our regret analysis, all of which are standard in the literature on neural bandits (Zhou et al., 2020; Zhang et al., 2021).

**Assumption 2.** *Without loss of generality, we assume the following:*

- *The score function is bounded: $|r^*(\mathbf{x})| \le 1$, for all $\mathbf{x} \in \mathcal{X}_t$, $t \in [T]$;*

- *There exists $\lambda_0 > 0$ such that the kernel matrix satisfies $\mathbf{H} \succeq \lambda_0 \mathbf{I}$;*

- *All context-arm feature vectors satisfy $\|\mathbf{x}\|_2 = 1$, and are symmetric in the sense that $x_j = x_{j+d/2}$ for all $\mathbf{x} \in \mathcal{X}_t$, $t \in [T]$.*

Note that the last assumption from Assumption 2 can be easily satisfied via a simple feature space transformation (Zhou et al., 2020).

**Definition 1.** *Let $\mathbf{H}' = \sum_{s=1}^T \sum_{(i,j) \in \mathcal{C}_K^2} \frac{1}{m} \mathbf{z}_i^j(s) \left( \mathbf{z}_i^j(s) \right)^\top$, where $\mathbf{z}_i^j(s) = \phi(\mathbf{x}_{s,i}) - \phi(\mathbf{x}_{s,j})$, and $\mathcal{C}_N^2$ denotes the set of all pairwise combinations of the $N$ arms. We define the effective dimension $\widetilde{d}$ as:*

$$\widetilde{d} = \log \det \left( \frac{\kappa_\mu}{\lambda} \mathbf{H}' + \mathbf{I} \right). \tag{9}$$

*which measures the complexity of the feature space induced by the neural network.*

This definition is consistent with Verma et al. (2025) and generalizes the effective dimension introduced in Zhou et al. (2020) by incorporating a larger set of pairwise comparisons and the scaling factor $\kappa_\mu/\lambda$, thus capturing the combinatorial structure of our dueling setting.

**Lemma 2.** *Let $\epsilon_{m,t}' = C_2 m^{-1/6} \sqrt{\log m} L^3 (\frac{t}{\lambda})^{4/3}$ for some absolute constant $C_2 > 0$ and $\delta \in (0, 1)$. As long as $m \ge poly(T, L, K, 1/\kappa_\mu, L_\mu, 1/\lambda_0, 1/\lambda, \log(1/\delta))$, Let the context of two super arms $S_t^1$ and $S_t^2$ to be $\mathcal{X}_t(S_t^1) = \{\mathbf{x}_{t,i}^1\}_{i=1}^k$, $\mathcal{X}_t(S_t^2) = \{\mathbf{x}_{t,i}^2\}_{i=1}^k$, then for all $t \in [T]$, with probability of at least $1 - \delta$, we have*

$$\left| \left( f(S_t^1, \mathbf{r}^*) - f(S_t^2, \mathbf{r}^*) \right) - \left( f(S_t^1, \mathbf{r}_t) - f(S_t^2, \mathbf{r}_t) \right) \right| \le \nu_T \sigma_{t-1}(S_t^1, S_t^2) + 2k\epsilon_{m,t}'.$$

Note that when the width of our neural network $m$ is large enough, which is satisfied by Eq. equation 34, we could make sure that $\epsilon_{m,t}' = C_2 m^{-1/6} \sqrt{\log m} L^3 (\frac{t}{\lambda})^{4/3}$ will be small enough, such that $\epsilon_{m,t}' = \widetilde{O}(1/T)$. Note that $f(S, \mathbf{r})$ is the reward function associated with super arm $S$ and score $\mathbf{r}$ of each single arm. Recall that $f(S, \mathbf{r}^*)$ is the reward of super arm $S$ with true score $\mathbf{r}^*$ and $f(S, \mathbf{r}_t)$ is the reward of super arm $S$ with estimated score $\mathbf{r}_t$ vector of each single arm where $r_t(\mathbf{x}) = h(\mathbf{x}; \theta_t)$ using neural network. Lemma 2 shows that the reward difference between two super arms $S_t^1$ and $S_t^2$ with the real score vector $\mathbf{r}^*$ and estimated score vector $\mathbf{r}_t$ is upper-bounded. That is, the estimation error of the reward difference using $\mathbf{r}_t$ is guaranteed to be small.

**Theorem 2** (NCDB). *Let* $\lambda \geq \kappa_\mu$, $B$ *be a constant such that* $\sqrt{2\mathbf{h}^\top \mathbf{H}^{-1}\mathbf{h}} \leq B$, $\beta_T \triangleq \frac{1}{\kappa_\mu}\sqrt{\widetilde{d} + 2\log(1/\delta)}$ *and* $c_0 > 0$ *be a constant such that* $\frac{1}{m}\|\phi(\mathbf{x}) - \phi(\mathbf{x}')\|_2^2 \leq c_0$ *for all* $\mathbf{x}, \mathbf{x}' \in \mathcal{X}_t$, $t \in [T]$ *and* $m \geq \text{poly}\left(T, L, K, \frac{1}{\kappa_\mu}, L_\mu, \frac{1}{\lambda_0}, \frac{1}{\lambda}, \log(1/\delta)\right)$, *then we have*

$$Reg_T \leq 3(\beta_T + B\sqrt{\frac{\lambda}{\kappa_\nu}} + 1)\sqrt{Tk2c_0\widetilde{d}} + 1 = \widetilde{O}\left(\left(\frac{1}{\kappa_\mu}\sqrt{\widetilde{d}} + B\sqrt{\frac{\lambda}{\kappa_\mu}}\right)\sqrt{Tk\widetilde{d}}\right). \quad (10)$$

If we follow the previous works (Bengs et al., 2022; Verma et al., 2025) and assume that $\widetilde{d} = \widetilde{o}(\sqrt{T})$, the regret upper bound of NCDB from Theorem 2 becomes sublinear. This condition reflects a mild growth constraint on the complexity of the feature space and is standard in the analysis of both linear and neural bandit algorithms.

Compared to the regret bound established for neural dueling bandits (Verma et al., 2025), our bound includes an additional dependence on the parameter $k$, which is natural since $k$ reflects the size of the super arm selected in each round under the combinatorial setting. In contrast to the regret bound of contextual neural bandits (Hwang et al., 2023), which achieves $\widetilde{O}\left(\sqrt{\widetilde{d}T\max(\widetilde{d}, k)}\right)$, our bound is relatively looser (e.g., it additionally depends on $1/\kappa_\mu$). This discrepancy arises from the more limited feedback (i.e., preference feedback) in the dueling bandit setting, where only pairwise preferences are observed rather than full reward information for each arm. We also provide an analysis of the regret under the Lipschitz continuity assumption in Appendix F.

## 5 EXPERIMENTS

We empirically evaluate the performance of our proposed algorithms in synthetic contextual environments. We consider a contextual environment where each arm is represented by a $d$-dimensional feature vector. Similar to (Zhou et al., 2020; Verma et al., 2025; Hwang et al., 2023) we adopt the following score functions: $r^*(\mathbf{x}) = \mathbf{x}^\top\theta$, $r^*(\mathbf{x}) = 10(\mathbf{x}^\top\theta)^2$ and $r^*(\mathbf{x}) = \cos(3\mathbf{x}^\top\theta)$. They correspond to linear, cosine, and square functions, and $\theta$ is a parameter to generate different score functions. The experimental settings are described in Appendix A.

We conduct a comparative evaluation between our proposed algorithm and a uniform sampling strategy, where super arms are sampled uniformly at random from the entire feasible set. Since the problem setting we study is novel and, to the best of our knowledge, has not been addressed in prior literature, there are no existing algorithms specifically designed for this setting. Moreover, standard baselines cannot be directly applied to this problem. Therefore, we use uniform sampling as a comparison baseline to demonstrate the effectiveness of our approach.

### 5.1 REGRET COMPARISONS

In our first experiment, we set the total number of arms to $N = 5$ and the context dimension of each arm to $d = 5$, a configuration commonly adopted in some prior dueling bandits studies (Verma et al., 2025). The size of each super arm is set to $k = 2$, and the total number of rounds is $T = 500$. The results are shown in Figure 1.

For the linear latent score function $r^*(\mathbf{x}) = \mathbf{x}^\top\theta$, as shown in Figures 1a and 1b, our LinCDB algorithm significantly outperforms both the uniform sampling baseline and the neural network-based algorithm NCDB. These results highlight the effectiveness of LinCDB when the underlying reward function is linear.

In contrast, for the nonlinear reward functions $r^*(\mathbf{x}) = \cos(3\mathbf{x}^\top\theta)$ and $r^*(\mathbf{x}) = 10(\mathbf{x}^\top\theta)^2$, the NCDB algorithm, which leverages a neural network to estimate the nonlinear score function, achieves the best performance. LinCDB performs poorly in these cases because it relies on linear approximation, which is not suitable for capturing the structure of cosine and quadratic functions.

These results suggest that when the reward function is known to be linear, LinCDB is the preferred choice. Otherwise, NCDB is more appropriate for handling nonlinear reward structures.

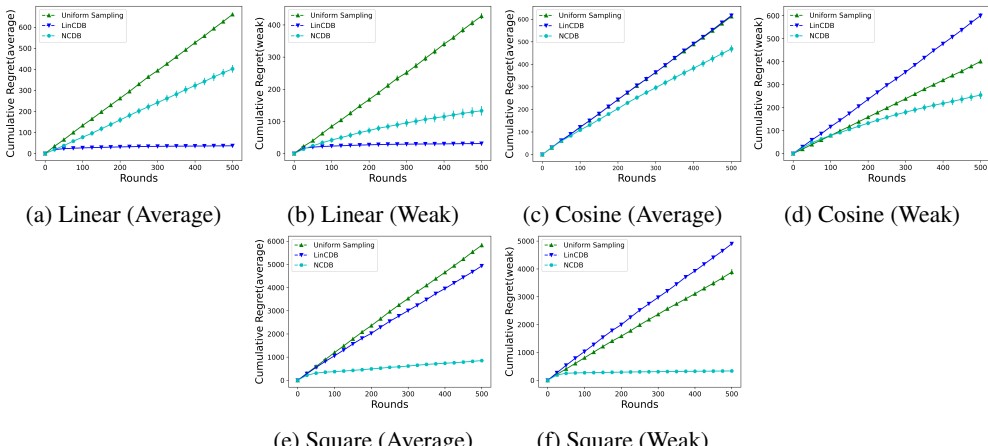

(a) Linear (Average)  (b) Linear (Weak)  (c) Cosine (Average)  (d) Cosine (Weak)

(e) Square (Average)  (f) Square (Weak)

Figure 1: Comparisons of cumulative regret (average and weak) of LinCDB, NCDB and Uniform Sampling algorithm on different score functions: Linear ($\mathbf{x}^\top \theta$), Cosine ($10(\mathbf{x}^\top \theta)^2$) and Square ($\cos(3\mathbf{x}^\top \theta)$).

## 5.2 VARYING CONTEXT DIMENSION AND SUPER ARM SIZE

We evaluate the performance of our NCDB algorithm under varying context dimensions and super arm sizes.

**Varying Context Dimension.** We fix the number of arms $N = 5$ and the super arm size $k = 2$, using the square function as the reward. The context dimension $d$ is varied in $\{5, 10, 15, 20, 25\}$, with all other parameters held fixed. The average and weak regret results are shown in Figure 2 (a) and (b), which demonstrate that although a larger $d$ results in worse regrets, our NCDB algorithm is consistently able to achieve sublinear regrets.

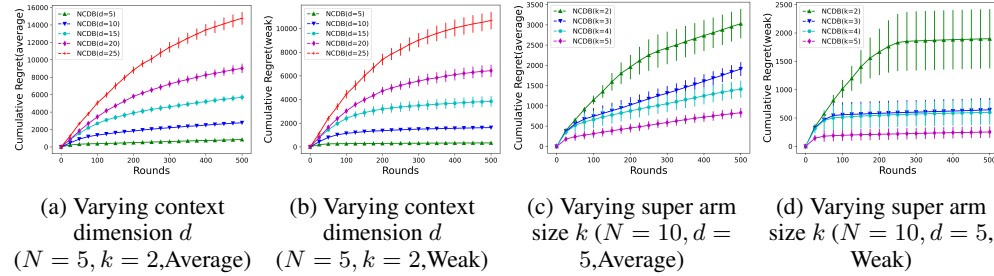

(a) Varying context dimension $d$ ($N = 5, k = 2$,Average)  (b) Varying context dimension $d$ ($N = 5, k = 2$,Weak)  (c) Varying super arm size $k$ ($N = 10, d = 5$,Average)  (d) Varying super arm size $k$ ($N = 10, d = 5$, Weak)

Figure 2: Comparisons of cumulative regret (average and weak) of NCDB algorithm as context dimension $d$ and super arm size $k$ increases.

**Varying Super Arm Size.** We fix $N = 10$ and $d = 5$, again using the square function. The super arm size $k$ is varied in $\{2, 3, 4, 5\}$, with other settings unchanged. Results are presented in Figure 2 (c) and (d), which show that a larger super arm size leads to smaller regrets. This is likely because a large super arm size leads to the availability of more observations.

## 6 CONCLUSION

We initiate the study of *combinatorial dueling bandits*, where the learner selects two super arms and receives feedback in the form of pairwise preferences between individual arms from the two super arms. We propose two algorithms: LinCDB for linear score functions and NCDB for nonlinear score functions. We theoretically show that LinCDB achieves a regret bound of $\widetilde{O}\left(\frac{d}{\kappa_\mu}\sqrt{Tk}\right)$, while NCDB achieves $\widetilde{O}\left(\left(\frac{1}{\kappa_\mu}\sqrt{\widetilde{d}} + B\sqrt{\frac{\lambda}{\kappa_\nu}}\right)\sqrt{Tk\widetilde{d}}\right)$.

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

## A EXPERIMENTAL SETTINGS

We generate a $d$-dimensional feature vector $x_{t,i} \in \mathbb{R}^d$ for each context-arm by sampling each entry independently and uniformly at random from the interval $(-1, 1)$. The ground-truth parameter vector $\theta \in \mathbb{R}^d$ is also sampled uniformly at random from the same interval and remains fixed throughout each run. Each experiment is repeated 20 times and we report both the weak cumulative regret and the average cumulative regret, along with 95% confidence intervals across the trials. The binary preference feedback between two super arms is simulated using a Bernoulli distribution with success probability $p = \mu(f(x_1) - f(x_2))$, where $\mu : \mathbb{R} \to [0, 1]$ is a link function. For all of the experiments, we fix the regularization parameter $\lambda = 1$ and and exploration variance $\nu_T = \nu = 1$.

To estimate the score function using neural network, we design a neural network with parameters $L = 2$ and $m = 50$, i.e., the neural network has depth 2 layers with width 50 and ReLU function is used as the activation function. Following prior work (Verma et al., 2025; Jacot et al., 2018), we use the current network parameters $\theta_t$ to compute the feature representations $g(x; \theta_t)$ at each round. Specifically, we recompute all $g(x; \theta_t)$ for the historical context-arm pairs whenever $\theta_t$ is updated. This ensures that the feature representations remain consistent with the most recent model state and allows the algorithm to better approximate the evolving reward structure during learning.

## B  RELATED WORK

**Stochastic Combinatorial Bandits.** Stochastic combinatorial bandits generalize the classical multi-armed bandit framework by allowing the selection of a super arm (i.e., a subset of base arms) rather than a single arm, and by receiving structured feedback accordingly. This setting has been extensively studied due to its wide applicability in real-world problems such as online advertising, recommendation systems, real-time vehicle routing, and network routing, where decisions naturally involve combinations of atomic actions (Chen et al., 2013; Wen et al., 2015; Gai et al., 2012; Qin et al., 2014; Atalar & Joe-Wong, 2025).

A key challenge in this domain is the exponentially large combinatorial action space. To address this, existing research has primarily adopted three modeling approaches. Early work often assumed a parametric structure, such as linear or generalized linear models (Qin et al., 2014; Wen et al., 2015; Kveton et al., 2015; Zong et al., 2016; Oh & Iyengar, 2019). More general approaches have relied on the Lipschitz continuity of the feedback model (Chen et al., 2018; Nika et al., 2020) or neural networks (Hwang et al., 2023) to handle complex nonlinearities.

Combinatorial Logistic Bandits (CLogB) further extend this line of work by modeling binary feedback using a logistic parametric model. This formulation is particularly suited for scenarios with complex arm-triggering structures and nonlinear reward functions (Liu et al., 2025). Although both our work and CLogB consider binary feedback, their feedback originates from independent arm outcomes rather than from dueling bandit comparisons, making the setting fundamentally different from ours.

In addition to assumptions on the feedback model, various assumptions have also been made regarding the reward function. Some studies adopt a simple additive model where the total reward is the sum of individual arm feedbacks (Wen et al., 2015; Kveton et al., 2015; Louëdec et al., 2015). Others generalize this to settings with submodular reward functions (Chen et al., 2018; Nika et al., 2020) or assume Lipschitz continuity of the reward function to accommodate more complex reward structures (Qin et al., 2014; Hwang et al., 2023).

**Contextual Dueling Bandits.** Dueling bandits have gained increasing attention in recent years (Yue & Joachims, 2009; 2011; Saha, 2021; Bengs et al., 2022; Zhu et al., 2023), where the learner selects a pair of arms and receives noisy binary feedback indicating the preference between them. The classical dueling bandit framework has been extended to the contextual linear stochastic transitivity model in (Bengs et al., 2022). To improve exploration efficiency, Di et al. (2023) proposed VACDB, an action-elimination based algorithm with a tighter variance-dependent regret bound. Li et al. (2024) introduced FGTS.CDB, a Thompson sampling algorithm tailored for linear contextual dueling bandits, which achieves a regret of $\mathcal{O}(d\sqrt{T})$. More recently, Verma et al. (2025) extended the contextual dueling bandit setting to nonlinear reward functions using neural networks, and developed UCB and Thompson sampling algorithms with sublinear regret guarantees.

## C  HUNGARIAN ALGORITHM

In our algorithm, the second super arm $S_t^2$ is selected by maximizing the UCB value.

Given the context set $\mathcal{X}_t = \{\mathbf{x}_{t,1}, \mathbf{x}_{t,2}, \ldots, \mathbf{x}_{t,N}\}$, the score vector $\mathbf{r}_t = [r_{t,i}]_{i=1}^N = [r_t(\mathbf{x}_{t,i})]_{i=1}^N$, and the first super arm $S_t^1 = \{s_{t,1}^1, s_{t,2}^1, \ldots, s_{t,k}^1\}$ and the context of the first super arm $\mathcal{X}_t(S_t^1) = \{\mathbf{x}_{t,1}^1, \mathbf{x}_{t,2}^1, \ldots, \mathbf{x}_{t,k}^1\}$, we aim to find the second super arm by solving the following problem:

$$S_t^2 = \arg\max_{S \in \mathcal{S}} \left[ f(S, \mathbf{r}_t) + \frac{\beta_t}{\kappa_\mu} \sigma_{\mathcal{X}_t}(S, S_t^1) \right]. \tag{11}$$

However, enumerating all possible super arms $S \in \mathcal{S}$ is computationally expensive due to the exponential size of the action space. To address this issue, we formulate the problem as a bipartite matching task and solve it using the Hungarian algorithm.

We decompose the objective function as follows:

$$f(S_t^2, \mathbf{r}_t) + \frac{\beta_t}{\kappa_\mu} \sigma_{\mathcal{X}_t}(S_t^2, S_t^1) = \sum_{i=1}^k \left( r_t(\mathbf{x}_{t,i}^2) + \frac{\beta_t}{\kappa_\mu} \left\| \mathbf{x}_{t,i}^1 - \mathbf{x}_{t,i}^2 \right\|_{V_{t-1}^{-1}} \right), \tag{12}$$

where $\mathbf{x}_{t,i}^1$ and $\mathbf{x}_{t,i}^2$ denote the context vectors of the corresponding arms in $S_t^1$ and $S_t^2$, respectively.

For each context $\mathbf{x}_{t,i}^1 \in S_t^1$ and each $\mathbf{x} \in \mathcal{X}_t$, we define the matching value as:

$$\text{value}(\mathbf{x}_{t,i}^1, \mathbf{x}) = r_t(\mathbf{x}) + \frac{\beta_t}{\kappa_\mu} \left\| \mathbf{x}_{t,i}^1 - \mathbf{x} \right\|_{V_{t-1}^{-1}}. \tag{13}$$

Therefore, the original maximization can be reformulated as:

$$S_t^2 = \arg\max_{S \in \mathcal{S}} \left( \sum_{\mathbf{x} \in \mathcal{X}_t(S)} \text{value}(\mathbf{x}_{t,i}^1, \mathbf{x}) \right), \tag{14}$$

which is equivalent to solving a maximum-weight bipartite matching problem, where each node in $S_t^1$ is matched with a node in $\mathcal{X}_t$. We present the procedure for selecting the second super arm $S_t^2$ in Algorithm 3, where the selection task is formulated as a bipartite matching problem and solved via the Hungarian Algorithm. Specifically, we construct a cost matrix based on the estimated reward vector and a confidence-adjusted distance metric between each element in the first super arm $S_t^1$ and the candidate context set $\mathcal{X}_t$. The detailed steps of the Hungarian Algorithm for solving the resulting assignment problem are outlined in Algorithm 4.

---

**Algorithm 3** Selecting Second Super Arm $S_t^2$ via Hungarian Algorithm

---

**Require:** First super arm $S_t^1 = \{\mathbf{x}_{t,1}^1, \ldots, \mathbf{x}_{t,k}^1\}$, context set $\mathcal{X}_t$, score vector $\mathbf{r}_t$, matrix $V_{t-1}$, parameters $\beta_t, \kappa_\mu$
**Ensure:** Second super arm $S_t^2$
1: Initialize cost matrix $C \in \mathbb{R}^{k \times N}$
2: **for** each $i = 1$ to $k$ **do**
3:     **for** each $j = 1$ to $N$ **do**
4:       $C_{i,j} \leftarrow - \left( r_t(\mathbf{x}_{t,j}) + \frac{\beta_t}{\kappa_\mu} \left\| \mathbf{x}_{t,i}^1 - \mathbf{x}_{t,j} \right\|_{V_{t-1}^{-1}} \right)$
5:     **end for**
6: **end for**
7: $M \leftarrow \text{HungarianAlgorithm}(C)$             // solve min-cost matching
8: $S_t^2 \leftarrow \{\mathbf{x}_{t,j} \mid (i,j) \in M\}$
9: **return** $S_t^2$

---

# D THEORETICAL ANALYSIS OF LINEAR COMBINATORIAL DUELING BANDITS

## D.1 PROOF OF LEMMA 1

**Lemma 1.** In any iteration $t = 1, \ldots, T$, for $\mathbf{x}_1, \mathbf{x}_2 \in \mathcal{X}_t$ with probability of at least $1 - \delta$, we have that

$$\left| (r^*(\mathbf{x}_1) - r^*(\mathbf{x}_2)) - (r_t(\mathbf{x}_1) - r_t(\mathbf{x}_2)) \right| \leq \frac{\beta_t}{\kappa_\mu} \left\| \mathbf{x}_1 - \mathbf{x}_2 \right\|_{V_{t-1}^{-1}}$$

**Lemma 3.** Let $\beta_t \triangleq \sqrt{2 \log(1/\delta) + d \log \left( 1 + tkD^2 \kappa_\mu / (d\lambda) \right)}$. For all $t = 1, \ldots, T$, With probability of at least $1 - \delta$, we have that

$$\left\| \theta - \theta_t \right\|_{V_t} \leq \frac{\beta_t}{\kappa_\mu}.$$

*Proof.* Define $\widetilde{\mathbf{x}}_{t,i} = \mathbf{x}_{t,i}^1 - \mathbf{x}_{t,i}^2$.

For any $\theta_{r'} \in \mathbb{R}^d$, define

$$G_t(\theta_{r'}) = \sum_{s=1}^t \sum_{i=1}^k \left( \mu(\theta_{r'}^\top \widetilde{\mathbf{x}}_{s,i}) - \mu(\theta^\top \widetilde{\mathbf{x}}_{s,i}) \right) \widetilde{\mathbf{x}}_{s,i} + \lambda \theta_{r'}.$$

---

**Algorithm 4** Hungarian Algorithm for Assignment Problem

---

**Require:** Cost matrix $C \in \mathbb{R}^{n \times n}$
**Ensure:** Optimal assignment $M$
1: Subtract row minimum from each row of $C$
2: **for** each row $i$ **do**
3:     $C_{i,:} \leftarrow C_{i,:} - \min_j C_{i,j}$
4: **end for**
5: Subtract column minimum from each column
6: **for** each column $j$ **do**
7:     $C_{:,j} \leftarrow C_{:,j} - \min_i C_{i,j}$
8: **end for**
9: Initialize cover lines and marked zeros
10: **repeat**
11:     Find a zero in $C$ and mark it if no other zero in its row/column is marked
12:     Cover all columns containing marked zeros
13:     **if** number of covered columns equals $n$ **then**
14:         **break**
15:     **else**
16:         Find the smallest uncovered value $h$
17:         Subtract $h$ from all uncovered elements
18:         Add $h$ to all elements covered twice
19:     **end if**
20: **until** all assignments are made
21: Construct optimal assignment $M$ from marked zeros
22: **return** $M$

---

Let $V_t = \sum_{s=1}^{t} \sum_{i=1}^{k} \widetilde{\mathbf{x}}_{s,i} \widetilde{\mathbf{x}}_{s,i}^{\top} + \frac{\lambda}{\kappa_\mu} \mathbf{I}$ . For $\lambda' \in (0,1)$, setting $\theta_{\bar{r}} = \lambda' \theta_{r_1'} + (1-\lambda') \theta_{r_2'}$ and using the mean-value theorem, we get:

$$
\begin{aligned}
G_t(\theta_{r_1'}) - G_t(\theta_{r_2'}) &= \left[ \sum_{s=1}^{t} \sum_{i=1}^{k} \dot{\mu}(\theta_{\bar{r}}^{\top} \widetilde{\mathbf{x}}_{s,i}) \widetilde{\mathbf{x}}_{s,i} \widetilde{\mathbf{x}}_{s,i}^{\top} + \lambda \mathbf{I} \right] (\theta_{r_1'} - \theta_{r_2'}) && (\theta \text{ is constant}) \\
&\geq \kappa_\mu \left[ \sum_{s=1}^{t} \sum_{i=1}^{k} \widetilde{\mathbf{x}}_{s,i} \widetilde{\mathbf{x}}_{s,i}^{\top} + \frac{\lambda}{\kappa_\mu} \mathbf{I} \right] (\theta_{r_1'} - \theta_{r_2'}) && \left(\text{from Assumption 1: } \dot{\mu}(\theta_{\bar{r}}^{\top} \widetilde{\mathbf{x}}_{s,i}) \geq \kappa_\mu \right) \\
&= \kappa_\mu V_t (\theta_{r_1'} - \theta_{r_2'}) && \left( \text{setting } V_t = \sum_{s=1}^{t} \sum_{i=1}^{k} \widetilde{\mathbf{x}}_{s,i} \widetilde{\mathbf{x}}_{s,i}^{\top} + \frac{\lambda}{\kappa_\mu} \mathbf{I} \right)
\end{aligned}
$$
$$(15)$$

Now using 15, we have that

$$
\begin{aligned}
\|G_t(\theta_t)\|_{V_t^{-1}}^2 &= \|G_t(\theta) - G_t(\theta_t)\|_{V_t^{-1}}^2 && (G_t(\theta) = 0 \text{ by definition}) \\
&\geq (\kappa_\mu V_t(\theta - \theta_t))^{\top} V_t^{-1} \kappa_\mu V_t(\theta - \theta_t) && (\text{as } \|x\|_A^2 = x^{\top} A x) \\
&= \kappa_\mu^2 (\theta - \theta_t)^{\top} V_t V_t^{-1} V_t (\theta - \theta_t) && (\text{as } V_t^{\top} = V_t \text{ and } \kappa_\mu \text{ is constant}) \\
&= \kappa_\mu^2 \|\theta - \theta_t\|_{V_t}^2 && (\text{as } \|x\|_A^2 = x^{\top} A x)
\end{aligned}
$$

Note that $\widetilde{\mathbf{x}}_{t,i} = \mathbf{x}_{t,i}^1 - \mathbf{x}_{t,i}^2$. Let $\widetilde{r}_{t,s,i} = r_t(\widetilde{\mathbf{x}}_{s,i}) = \theta_t^\top \widetilde{\mathbf{x}}_{s,i}$ and $\widetilde{r}_{s,i}^* = r^*(\widetilde{\mathbf{x}}_{s,i}) = \theta^\top \widetilde{\mathbf{x}}_{s,i}$, in which $\theta_t$ is our empirical estimate of the unknown parameter $\theta$. This allows us to show that:

$$\|\theta - \theta_t\|_{V_t}^2 \leq \frac{1}{\kappa_\mu^2} \|G_t(\theta_t)\|_{V_t^{-1}}^2$$

$$= \frac{1}{\kappa_\mu^2} \left\| \sum_{s=1}^t \sum_{i=1}^k (\mu(\theta_t^\top \widetilde{\mathbf{x}}_{s,i}) - \mu(\theta^\top \widetilde{\mathbf{x}}_{s,i}))\widetilde{\mathbf{x}}_{s,i} + \lambda\theta_t \right\|_{V_t^{-1}}^2 \qquad \text{(by definition of } G_t(\theta_t))$$

$$= \frac{1}{\kappa_\mu^2} \left\| \sum_{s=1}^t \sum_{i=1}^k (\mu(\widetilde{r}_{t,s,i}) - \mu(\widetilde{r}_{s,i}^*))\widetilde{\mathbf{x}}_{s,i} + \lambda\theta_t \right\|_{V_t^{-1}}^2 \qquad \text{(see definitions of } \widetilde{r}_{t,s,i} \text{ and } \widetilde{r}_{s,i})$$

$$= \frac{1}{\kappa_\mu^2} \left\| \sum_{s=1}^t \sum_{i=1}^k (\mu(\widetilde{r}_{t,s,i}) - (y_{s,i} - \epsilon_{s,i}))\widetilde{\mathbf{x}}_{s,i} + \lambda\theta_t \right\|_{V_t^{-1}}^2 \qquad \text{(as } y_{s,i} = \mu(\widetilde{r}_{s,i}^*) + \xi_{s,i})$$

$$= \frac{1}{\kappa_\mu^2} \left\| \sum_{s=1}^t \sum_{i=1}^k (\mu(\widetilde{r}_{t,s,i}) - y_{s,i})\widetilde{\mathbf{x}}_{s,i} + \sum_{s=1}^t \sum_{i=1}^k \xi_{s,i}\widetilde{\mathbf{x}}_{s,i} + \lambda\theta_t \right\|_{V_t^{-1}}^2$$

$$\leq \frac{1}{\kappa_\mu^2} \left\| \sum_{s=1}^t \sum_{i=1}^k \xi_{s,i}\widetilde{\mathbf{x}}_{s,i} \right\|_{V_t^{-1}}^2.$$

The last step follows from the fact that $\theta_t$ is computed using MLE by solving the following equation:

$$\sum_{s=1}^t \sum_{i=1}^k \left( \mu\left(\theta_t^\top \widetilde{\mathbf{x}}_{s,i}\right) - y_{s,i} \right) \widetilde{\mathbf{x}}_{s,i} + \lambda\theta_t = 0, \tag{16}$$

which is ensured by equation 2.

With this, we have

$$\|\theta - \theta_t\|_{V_t}^2 \leq \frac{1}{\kappa_\mu^2} \left\| \sum_{s=1}^t \sum_{i=1}^k \xi_{s,i}\widetilde{\mathbf{x}}_{s,i} \right\|_{V_t^{-1}}^2 \tag{17}$$

Denote the observation noise $\xi_{s,i} = y_{s,i} - \mu(r(\mathbf{x}_{s,i}^1) - r(\mathbf{x}_{s,i}^2))$. Note that the sequence of observation noises $\{\xi_{s,i}\}$ is 1-sub-Gaussian and $V_t = \sum_{s=1}^t \sum_{i=1}^k \widetilde{\mathbf{x}}_{s,i}\widetilde{\mathbf{x}}_{s,i}^\top + \frac{\lambda}{\kappa_\mu}\mathbf{I}$.

Next, we can apply Theorem 1 from (Abbasi-Yadkori et al., 2011), to obtain

$$\left\| \sum_{s=1}^t \sum_{i=1}^k \xi_{s,i}\widetilde{\mathbf{x}}_{s,i} \right\|_{V_t^{-1}}^2 \leq 2\log\left( \frac{\det(V_t)^{1/2}}{\delta \det(V)^{1/2}} \right), \tag{18}$$

which holds with probability of at least $1 - \delta$.

Then, based on our assumption that $\|\widetilde{\mathbf{x}}_{s,i}\|_2 \leq D$ (Assumption 1), according to Lemma 10 from (Abbasi-Yadkori et al., 2011), we have that

$$\det(V_t) \leq (\lambda/\kappa_\mu + tkD^2/d)^d. \tag{19}$$

Therefore,

$$\sqrt{\frac{\det V_t}{\det(V)}} \leq \sqrt{\frac{(\lambda/\kappa_\mu + tkD^2/d)^d}{(\lambda/\kappa_\mu)^d}} = \left(1 + tkD^2\kappa_\mu/(d\lambda)\right)^{\frac{d}{2}} \tag{20}$$

This gives us

$$\left\| \sum_{s=1}^t \sum_{i=1}^k \epsilon_{s,i}\widetilde{\mathbf{x}}_{s,i} \right\|_{V_t^{-1}}^2 \leq 2\log\left( \frac{\det(V_t)^{1/2}}{\delta \det(V)^{1/2}} \right) \leq 2\log(1/\delta) + d\log\left(1 + tkD^2\kappa_\mu/(d\lambda)\right) \tag{21}$$

Combining equation 17 and equation 21, we have that

$$\|\theta - \theta_t\|_{V_t}^2 \leq \frac{1}{\kappa_\mu^2} \left( 2\log(1/\delta) + d\log\left(1 + tkD^2\kappa_\mu/(d\lambda)\right)\right) = \frac{\beta_t^2}{\kappa_\mu^2}, \tag{22}$$

which completes the proof.

$\square$

**Lemma 1.** In any iteration $t = 1, \ldots, T$, for $\mathbf{x}_1, \mathbf{x}_2 \in \mathcal{X}_t$ with probability of at least $1 - \delta$, we have that

$$|\left(r^*(\mathbf{x}_1) - r^*(\mathbf{x}_2)\right) - \left(r_t(\mathbf{x}_1) - r_t(\mathbf{x}_2)\right)| \leq \frac{\beta_t}{\kappa_\mu}\|\mathbf{x}_1 - \mathbf{x}_2\|_{V_{t-1}^{-1}}$$

*Proof.*

$$\begin{aligned}
|\left(r^*(\mathbf{x}_1) - r^*(\mathbf{x}_2)\right) - \theta_t^\top\left(\mathbf{x}_1 - \mathbf{x}_2\right)| &= |\theta^\top\left[(\mathbf{x}_1 - \mathbf{x}_2\right] - \theta_t^\top\left[\mathbf{x}_1 - \mathbf{x}_2\right]|\\
&= |\left(\theta - \theta_t\right)^\top\left[\mathbf{x}_1 - \mathbf{x}_2\right]|\\
&\leq \|\theta - \theta_t\|_{V_{t-1}}\|\mathbf{x}_1 - \mathbf{x}_2\|_{V_{t-1}^{-1}} \quad\quad (23)\\
&\leq \frac{\beta_t}{\kappa_\mu}\|\mathbf{x}_1 - \mathbf{x}_2\|_{V_{t-1}^{-1}},
\end{aligned}$$

in which the last inequality follows from Lemma 3.

$\square$

Lemma 1 immediately tells us that

$$r_t(\mathbf{x}_1) - r_t(\mathbf{x}_2) \leq r^*(\mathbf{x}_1) - r^*(\mathbf{x}_2) + \frac{\beta_t}{\kappa_\mu}\|\mathbf{x}_1 - \mathbf{x}_2\|_{V_{t-1}^{-1}}$$

$$\tag{24}$$

$$r_t(\mathbf{x}_1) - r_t(\mathbf{x}_2) + \frac{\beta_t}{\kappa_\mu}\|\mathbf{x}_1 - \mathbf{x}_2\|_{V_{t-1}^{-1}} \leq r^*(\mathbf{x}_1) - r^*(\mathbf{x}_2) + 2\frac{\beta_t}{\kappa_\mu}\|\mathbf{x}_1 - \mathbf{x}_2\|_{V_{t-1}^{-1}}.$$

## D.2 PROOF OF THEOREM 1

**Lemma 4.** *For any two super arm $S_t^1, S_t^2 \in \mathcal{S}$, Define*

$$\sigma_{\mathcal{X}_t}(S_t^1, S_t^2) \triangleq \sum_{i=1}^{k}\left\|\mathbf{x}_{t,i}^1 - \mathbf{x}_{t,i}^2\right\|_{V_{t-1}^{-1}}.$$

*Then we have that*

$$\left|\left(f(S_t^1, \mathbf{r}^*) - f(S_t^2, \mathbf{r}^*)\right) - \left(f(S_t^1, \mathbf{r}_t) - f(S_t^2, \mathbf{r}_t)\right)\right| \leq \frac{\beta_t}{\kappa_\mu}\sigma_{\mathcal{X}_t}(S_t^1, S_t^2).$$

*Proof.* Following Assumption 1, we have that

$$
\begin{aligned}
& \left| \left( f(S_t^1, \mathbf{r}^*) - f(S_t^2, \mathbf{r}^*) \right) - \left( f(S_t^1, \mathbf{r}_t) - f(S_t^2, \mathbf{r}_t) \right) \right| \\
&= \left| \left[ \sum_{i=1}^k r^*(\mathbf{x}_{t,i}^1) - \sum_{i=1}^k r^*(\mathbf{x}_{t,i}^2) - \sum_{i=1}^k r_t(\mathbf{x}_{t,i}^1) + \sum_{i=1}^k r_t(\mathbf{x}_{t,i}^2) \right] \right| \\
&= \left| \sum_{i=1}^k \left[ \left( r^*(\mathbf{x}_{t,i}^1) - r^*(\mathbf{x}_{t,i}^2) \right) - \left( r_t(\mathbf{x}_{t,i}^1) - r_t(\mathbf{x}_{t,i}^2) \right) \right] \right| \\
&\leq \sum_{i=1}^k \left| \left[ \left( r^*(\mathbf{x}_{t,i}^1) - r^*(\mathbf{x}_{t,i}^2) \right) - \left( r_t(\mathbf{x}_{t,i}^1) - r_t(\mathbf{x}_{t,i}^2) \right) \right] \right| \\
&\leq \frac{\beta_t}{\kappa_\mu} \sum_{i=1}^k \left\| \mathbf{x}_{t,i}^1 - \mathbf{x}_{t,i}^2 \right\|_{V_{t-1}^{-1}} \\
&= \frac{\beta_t}{\kappa_\mu} \sigma_{\mathcal{X}_t}(S_t^1, S_t^2).
\end{aligned}
\tag{25}
$$

The first equality holds by the definition of the reward function $f$.

$\square$

**Lemma 5.** *In any iteration $t$, the regret is bound by*

$$
reg_t \leq 3 \frac{\beta_t}{\kappa_\mu} \sum_{i=1}^k \left\| \mathbf{x}_{t,i}^1 - \mathbf{x}_{t,i}^2 \right\|_{V_{t-1}^{-1}}
$$

*Proof.* To begin with, for three super arms $S_t^1, S_t^2, S_t^*$, according to the definition of $\sigma_{\mathcal{X}_t}(\cdot, \cdot)$ Lemma 4, we have that

$$
\begin{aligned}
\sigma_{\mathcal{X}_t}(S_t^*, S_t^1) + \sigma_{\mathcal{X}_t}(S_t^1, S_t^2) &= \sum_{i=1}^k \left\| \mathbf{x}_{t,i}^* - \mathbf{x}_{t,i}^1 \right\|_{V_{t-1}^{-1}} + \sum_{i=1}^k \left\| \mathbf{x}_{t,i}^1 - \mathbf{x}_{t,i}^2 \right\|_{V_{t-1}^{-1}} \\
&\leq \sum_{i=1}^k \left\| \mathbf{x}_{t,i}^* - \mathbf{x}_{t,i}^2 \right\|_{V_{t-1}^{-1}} \\
&= \sigma_{\mathcal{X}_t}(S_t^*, S_t^2)
\end{aligned}
\tag{26}
$$

which follows from the triangle inequality.

That is

$$
\sigma_{\mathcal{X}_t}(S_t^*, S_t^2) \leq \sigma_{\mathcal{X}_t}(S_t^*, S_t^1) + \sigma_{\mathcal{X}_t}(S_t^1, S_t^2)
\tag{27}
$$

$$reg_t = 2\text{opt}_{\mathbf{r}^*} - \left(f(S_t^1, \mathbf{r}^*) + f(S_t^2, \mathbf{r}^*)\right)$$

$$= f(S_t^*, \mathbf{r}^*) - f(S_t^1, \mathbf{r}^*) + f(S_t^*, \mathbf{r}^*) - f(S_t^2, \mathbf{r}^*)$$

$$\overset{(a)}{\leq} f(S_t^*, \mathbf{r}_t) - f(S_t^1, \mathbf{r}_t) + \frac{\beta_t}{\kappa_\mu}\sigma_{\mathcal{X}_t}(S_t^*, S_t^1) + f(S_t^*, \mathbf{r}_t) - f(S_t^2, \mathbf{r}_t) + \frac{\beta_t}{\kappa_\mu}\sigma_{\mathcal{X}_t}(S_t^*, S_t^2)$$

$$\overset{(b)}{\leq} f(S_t^*, \mathbf{r}_t) - f(S_t^1, \mathbf{r}_t) + \frac{\beta_t}{\kappa_\mu}\sigma_{\mathcal{X}_t}(S_t^*, S_t^1) +$$

$$f(S_t^*, \mathbf{r}_t) - f(S_t^1, \mathbf{r}_t) + f(S_t^1, \mathbf{r}_t) - f(S_t^2, \mathbf{r}_t) + \frac{\beta_t}{\kappa_\mu}\sigma_{\mathcal{X}_t}(S_t^*, S_t^1) + \frac{\beta_t}{\kappa_\mu}\sigma_{\mathcal{X}_t}(S_t^1, S_t^2)$$

$$= 2\left(f(S_t^*, \mathbf{r}_t) - f(S_t^1, \mathbf{r}_t) + \frac{\beta_t}{\kappa_\mu}\sigma_{\mathcal{X}_t}(S_t^*, S_t^1)\right) + f(S_t^1, \mathbf{r}_t) - f(S_t^2, \mathbf{r}_t) + \frac{\beta_t}{\kappa_\mu}\sigma_{\mathcal{X}_t}(S_t^1, S_t^2)$$

$$\overset{(c)}{\leq} 2\left(f(S_t^2, \mathbf{r}_t) - f(S_t^1, \mathbf{r}_t) + \frac{\beta_t}{\kappa_\mu}\sigma_{\mathcal{X}_t}(S_t^2, S_t^1)\right) + f(S_t^1, \mathbf{r}_t) - f(S_t^2, \mathbf{r}_t) + \frac{\beta_t}{\kappa_\mu}\sigma_{\mathcal{X}_t}(S_t^1, S_t^2)$$

$$= f(S_t^2, \mathbf{r}_t) - f(S_t^1, \mathbf{r}_t) + 3\frac{\beta_t}{\kappa_\mu}\sigma_{\mathcal{X}_t}(S_t^1, S_t^2)$$

$$\overset{(d)}{\leq} 3\frac{\beta_t}{\kappa_\mu}\sigma_{\mathcal{X}_t}(S_t^1, S_t^2)$$

$$= 3\frac{\beta_t}{\kappa_\mu}\sum_{i=1}^k \left\|\mathbf{x}_{t,i}^1 - \mathbf{x}_{t,i}^2\right\|_{V_{t-1}^{-1}}$$

$$(28)$$

Inequality (a) follows from Lemma 4, inequality (b) makes use of equation 27. Inequality (c) follows from the way in which $S_t^2$ is selected:

$$S_t^2 = \arg\max_{S \in \mathcal{S}} f(S, \mathbf{r}_t) + \frac{\beta_t}{\kappa_\mu}\sigma_{\mathcal{X}_t}(S, S_t^1).$$

Note that $S_t^2$ is selected by Hungarian Algorithm in Appendix C. Inequality (d) results from the way in which $S_t^1$ is selected:

$$S_t^1 = \arg\max_{S \in \mathcal{S}} f(S, \mathbf{r}_t).$$

$\square$

**Lemma 6.**

$$\sum_{t=1}^T \sum_{i=1}^k \left\|\mathbf{x}_{t,i}^1 - \mathbf{x}_{t,i}^2\right\|_{V_{t-1}^{-1}} \leq \sqrt{2Tkd\log(1 + \frac{\kappa_\mu TkD^2}{d\lambda})}$$

*Proof.* First, we could show that

$$\det(V_t) = \det\left(V_{t-1} + \sum_{i=1}^k \widetilde{\mathbf{x}}_{t,i}\widetilde{\mathbf{x}}_{t,i}^\top\right)$$

$$= \det(V_{t-1})\det\left(I + \sum_{i=1}^k \left(V_{t-1}^{-\frac{1}{2}}\widetilde{\mathbf{x}}_{t,i}\right)\left(V_{t-1}^{-\frac{1}{2}}\widetilde{\mathbf{x}}_{t,i}\right)^T\right)$$

$$= \det(V_{t-1})\left(1 + \sum_{i=1}^k \left\|\mathbf{x}_{t,i}^1 - \mathbf{x}_{t,i}^2\right\|_{V_{t-1}^{-1}}^2\right)$$

$$= \det(V)\prod_{\tau=1}^t \left(1 + \sum_{i=1}^k \left\|\mathbf{x}_{\tau,i}^1 - \mathbf{x}_{\tau,i}^2\right\|_{V_{\tau-1}^{-1}}^2\right)$$

$$(29)$$

So that we have

$$\frac{\det(V_t)}{\det(V)} = \prod_{\tau=1}^{t} \left( 1 + \sum_{i=1}^{k} \left\| \mathbf{x}_{\tau,i}^1 - \mathbf{x}_{\tau,i}^2 \right\|_{V_{\tau-1}^{-1}}^2 \right) \tag{30}$$

According to our assumption 1, we have that for any $\mathbf{x}, \mathbf{x}' \in \mathcal{X}_t$, we have $\|\mathbf{x} - \mathbf{x}'\|_2 \leq D$. We use $V_{t-1}$ to denote the covariance matrix in $t$-th iteration, and denote $V = V_0 = \frac{\lambda}{\kappa_\mu} I$. It is easy to verify that $V_{t-1} \succeq \frac{\lambda}{\kappa_\mu} I$ and hence $V_{t-1}^{-1} \preceq \frac{\kappa_\mu}{\lambda} I$. Therefore, we have that $\left\| \mathbf{x}_{t,i}^1 - \mathbf{x}_{t,i}^2 \right\|_{V_{t-1}^{-1}}^2 \leq \frac{\kappa_\mu}{\lambda} \left\| \mathbf{x}_{t,i}^1 - \mathbf{x}_{t,i}^2 \right\|_2^2 \leq D^2 \frac{\kappa_\mu}{\lambda}$. We choose $\lambda$ such that $kD^2 \frac{\kappa_\mu}{\lambda} \leq 1$ , which ensures that $\sum_{i=1}^{k} \left\| \mathbf{x}_{t,i}^1 - \mathbf{x}_{t,i}^2 \right\|_{V_{t-1}^{-1}}^2 \leq 1$. Note that $x \leq 2\log(1+x)$ for $x \in [0,1]$. Then we have that

$$\begin{aligned}
\sum_{t=1}^{T} \sum_{i=1}^{k} \left\| \mathbf{x}_{t,i}^1 - \mathbf{x}_{t,i}^2 \right\|_{V_{t-1}^{-1}}^2 &\overset{(a)}{\leq} 2 \sum_{t=1}^{T} \log \left( 1 + \sum_{i=1}^{k} \left\| \mathbf{x}_{t,i}^1 - \mathbf{x}_{t,i}^2 \right\|_{V_{t-1}^{-1}}^2 \right) \\
&= 2 \sum_{t=1}^{T} \log \left( 1 + \sum_{i=1}^{k} \left\| \mathbf{x}_{t,i}^1 - \mathbf{x}_{t,i}^2 \right\|_{V_{t-1}^{-1}}^2 \right) \\
&\overset{(b)}{=} 2 \log(\frac{\det V_T}{\det V}) \\
&\overset{(c)}{\leq} 2 \log \left( (1 + \frac{\kappa_\mu T k D^2}{d\lambda})^d \right) \\
&= 2d \log(1 + \frac{\kappa_\mu T k D^2}{d\lambda})
\end{aligned} \tag{31}$$

where (a) follows from the fact that $x \leq 2\log(1+x)$ for $x \in [0,1]$ and the (b) follows from Eq.equation 30 and (c) follows from Lemma 10 from (Abbasi-Yadkori et al., 2011) that $\det(V_t) \leq (\lambda/\kappa_\mu + tkD^2/d)^d$.

And using Cauchy–Schwarz inequality, we can get

$$\begin{aligned}
\sum_{t=1}^{T} \sum_{i=1}^{k} \left\| \mathbf{x}_{t,i}^1 - \mathbf{x}_{t,i}^2 \right\|_{V_{t-1}^{-1}} &\leq \sqrt{Tk} \sqrt{\sum_{i=1}^{k} \sum_{t=1}^{T} \left\| \mathbf{x}_{t,i}^1 - \mathbf{x}_{t,i}^2 \right\|_{V_{t-1}^{-1}}^2} \\
&\leq \sqrt{2Tkd \log(1 + \frac{\kappa_\mu T k D^2}{d\lambda})}
\end{aligned} \tag{32}$$

$\square$

**Theorem 1.** Let $\beta_t \triangleq \sqrt{2\log(1/\delta) + d\log\left(1 + tkD^2\kappa_\mu/(d\lambda)\right)}$ and $\lambda \leq \frac{\kappa_\mu}{D^2}$, then With probability of at least $1 - \delta$, we have that

$$Reg_T \leq \frac{3}{\kappa_\mu} \sqrt{2\log(1/\delta) + d\log\left(1 + TkD^2\kappa_\mu/(d\lambda)\right)} \sqrt{2Tkd \log(1 + TkD^2\kappa_\mu/(d\lambda))}$$

*Proof.* Note that

$$r_t \leq 3\frac{\beta_t}{\kappa_\mu} \sum_{i=1}^{k} \left\| \mathbf{x}_{t,i}^1 - \mathbf{x}_{t,i}^2 \right\|_{V_{t-1}^{-1}}$$

So we have

$$\begin{aligned}
Reg_T \leq \sum_{t=1}^{T} r_t &\leq \sum_{t=1}^{T} 3\frac{\beta_t}{\kappa_\mu} \sum_{i=1}^{k} \left\| \mathbf{x}_{t,i}^1 - \mathbf{x}_{t,i}^2 \right\|_{V_{t-1}^{-1}} \\
&\leq 3\frac{\beta_T}{\kappa_\mu} \sqrt{2Tkd \log(1 + \frac{\kappa_\mu T k D^2}{d\lambda})}
\end{aligned} \tag{33}$$

So we have that $Reg_T \leq \frac{3}{\kappa_\mu} \sqrt{2 \log(1/\delta) + d \log\left(1 + TkD^2\kappa_\mu/(d\lambda)\right)} \sqrt{2Tkd \log(1 + TkD^2\kappa_\mu/(d\lambda))}$

$\square$

# E    THEORETICAL ANALYSIS OF NCDB

**Definition 2.** *(Zhou et al., 2020; Zhang et al., 2021) Let $\{\mathbf{x}^{(n)}\}_{n=1}^{TN}$ be the set of all possible context-arm feature vectors, i.e., $\{\mathbf{x}_{t,a}\}_{1 \leq t \leq T, \, 1 \leq a \leq N}$, where the index $n = N(t-1) + a$. Define the kernel recursively as follows:*

$$\mathbf{H}_{p,q}^{(1)} = \mathbf{\Sigma}_{p,q}^{(1)} = \langle \mathbf{x}^{(p)}, \mathbf{x}^{(q)} \rangle,$$

$$\mathbf{\Sigma}_{p,q}^{(\ell+1)} = 2 \cdot \mathbb{E}_{(u,v) \sim \mathcal{N}(0, A_{p,q}^{(\ell)})} \left[ \max\{u, 0\} \cdot \max\{v, 0\} \right],$$

$$\mathbf{H}_{p,q}^{(\ell+1)} = 2\mathbf{H}_{p,q}^{(\ell)} \cdot \mathbb{E}_{(u,v) \sim \mathcal{N}(0, A_{p,q}^{(\ell)})} \left[ \mathbb{I}(u \geq 0) \cdot \mathbb{I}(v \geq 0) \right] + \mathbf{\Sigma}_{p,q}^{(\ell+1)},$$

*where $A_{p,q}^{(\ell)} = \begin{bmatrix} \mathbf{\Sigma}_{p,p}^{(\ell)} & \mathbf{\Sigma}_{p,q}^{(\ell)} \\ \mathbf{\Sigma}_{q,p}^{(\ell)} & \mathbf{\Sigma}_{q,q}^{(\ell)} \end{bmatrix}$.*

*Finally, the matrix $\mathbf{H} = \frac{1}{2} \left( \mathbf{H}^{(L)} + \mathbf{\Sigma}^{(L)} \right)$ is called the* neural tangent kernel (NTK) *matrix over the set of context-arm feature vectors $\{\mathbf{x}^{(n)}\}_{n=1}^{TN}$.*

**Condition 1.** *Conditions needed for the width $m$ of NN:*

$$\begin{aligned}
m &\geq CT^4 N^4 L^6 \log(T^2 N^2 L/\delta)/\lambda_0^4, \\
m(\log m)^{-3} &\geq C\kappa_\mu^{-3} T^8 L^{21} \lambda^{-5}, \\
m(\log m)^{-3} &\geq C\kappa_\mu^{-3} T^{14} L^{21} \lambda^{-11} L_\mu^6, \\
m(\log m)^{-3} &\geq CT^{14} L^{18} \lambda^{-8},
\end{aligned} \tag{34}$$

*where $C$ is a positive absolute constant, $N$ is the number of available arms in each round. To simplify exposition, we can express the technical conditions above compactly as $m \geq \mathrm{poly}\left(T, L, N, 1/\kappa_\mu, L_\mu, 1/\lambda_0, 1/\lambda, \log(1/\delta)\right)$,*

For clarity, we adopt a unified error probability $\delta$ for all probabilistic statements throughout the analysis.

## E.1    ANALYSIS OF NEURAL NETWORK

**Lemma 7.** *(Lemma B.3 in (Zhang et al., 2021) ) As long as the width m of the NN is wide enough:*

$$m \geq C_0 T^4 K^4 L^6 \log(T^2 K^2 L/\delta)/\lambda_0^4$$

*then with probability of at least $1 - \delta$, there exits a $\theta_{r^*}$ such that*

$$r^*(\mathbf{x}) = \langle g(\mathbf{x}; \theta_0), \theta_{r^*} - \theta_0 \rangle$$

$$\sqrt{m} \, \|\theta_{r^*} - \theta_0\|_2 \leq \sqrt{2\mathbf{h}^\top \mathbf{H}^{-1} \mathbf{h}} \leq B$$

*for all $\mathbf{x} \in \mathcal{X}_t, \forall t \in [T]$*

**Lemma 8.** *We have that $\|\theta_t - \theta_0\|_2 \leq 2\sqrt{\frac{tk}{m\lambda}}, \forall t \in [T]$*

*Proof.* $\mu() \in [0, 1]$. Using Eq. 1 gives us

$$\frac{1}{2}\lambda \|\theta_t - \theta_0\|_2^2 \leq \mathcal{L}_t(\theta_t) \leq \mathcal{L}_t(\theta_0)$$

$$\overset{(a)}{=} -\frac{1}{m} \sum_{s=1}^{t-1} \sum_{i=1}^{k} (y_{s,i} \log \mu \left[ h(\mathbf{x}_{s,i}^1; \theta_0) - h(\mathbf{x}_{s,i}^2; \theta_0) \right]$$

$$+ (1 - y_{s,i}) \log \mu \left[ h(\mathbf{x}_{s,i}^2; \theta_0) - h(\mathbf{x}_{s,i}^1; \theta_0) \right]) + \frac{1}{2}\lambda \|\theta_0 - \theta_0\|_2^2.$$

$$= -\frac{1}{m} \sum_{s=1}^{t-1} \sum_{i=1}^{k} \left[ y_{s,i} \log \mu(0) + (1 - y_{s,i}) \log \mu(0) \right]$$

$$= -\frac{1}{m} \sum_{s=1}^{t-1} \sum_{i=1}^{k} \log 0.5$$

$$\leq \frac{tk}{m}(-\log 0.5)$$

$$\overset{(b)}{\leq} \frac{tk}{m}$$

Step $(a)$ follow because $h(\mathbf{x}; \theta_0) = 0, \forall \mathbf{x} \in \mathcal{X}_t, t \in [T]$ which is ensured by Assumption 2, step $(b)$ follows because $-\log 0.5 \leq 1$. Therefore, we have that $\|\theta_t - \theta_0\|_2 \leq \sqrt{2\frac{tk}{m\lambda}} \leq 2\sqrt{\frac{tk}{m\lambda}}$ □

**Lemma 9.** *Let* $\tau = 2\sqrt{\frac{tk}{m\lambda}}$. *Then for absolute constants* $C_3, C_1 \leq 0$, *with probability of at least* $1 - \delta$,

$$\|g(\mathbf{x}; \theta_t)\|_2 \leq C_3\sqrt{mL},$$

$$\|g(\mathbf{x}; \theta_0) - g(\mathbf{x}; \theta_t)\|_2 \leq C_1\sqrt{m \log m}\tau^{1/3}L^{7/2} = C_1 m^{1/3}\sqrt{\log m}(\frac{t}{\lambda})^{1/3}L^{7/2},$$

*for all* $\mathbf{x} \in \mathcal{X}_t, t \in [T]$.

*Proof.* It can be readily verified that our choice of $\tau = 2\sqrt{\frac{tk}{m\lambda}}$ satisfies the condition on $\tau$ required in Lemmas C.3 and C.4 of (Zhang et al., 2021). Consequently, we are able to invoke those results, as this choice ensures that $\|\theta_t - \theta_0\|_2 \leq \tau$. □

Lemma B.4 from (Zhou et al., 2020) allows us to obtain the following lemma, which shows that the output of NN can be approximated by its linearization.

**Lemma 10.** *((Zhou et al., 2020)) Let* $\tau \triangleq 2\sqrt{\frac{tk}{m\lambda}}$. *Let* $\epsilon'_{m,t} \triangleq C_2 m^{-1/6}\sqrt{\log m}L^3(\frac{t}{\lambda})^{4/3}$. *Then for some absolute constant* $C_2 \leq 0$, *with probability of at least* $1 - \delta$,

$$|h(\mathbf{x}; \theta_t) - \langle \theta_t - \theta_0, g(\mathbf{x}; \theta_0) \rangle| \leq C_2 \tau^{4/3}L^3\sqrt{m \log m} = C_2 m^{-1/6}\sqrt{\log m}L^3(\frac{t}{\lambda})^{4/3} = \epsilon'_{m,t}$$

*for all* $\mathbf{x} \in \mathcal{X}_t, t \in [T]$.

**Lemma 11.** *For all* $\mathbf{x}, \mathbf{x}' \in \mathcal{X}_t, t \in [T]$, *we have*

$$|\langle \phi(\mathbf{x}) - \phi(\mathbf{x}'), \theta_t - \theta_0 \rangle - (h(\mathbf{x}; \theta_t) - h(\mathbf{x}'; \theta_t))| \leq 2\epsilon'_{m,t}.$$

*Proof.* By re-arranging the left-hand side and then using Lemma 10, we get

$$|\langle \phi(\mathbf{x}) - \phi(\mathbf{x}'), \theta_t - \theta_0 \rangle - (h(\mathbf{x}; \theta_t) - h(\mathbf{x}'; \theta_t))|$$

$$= |\langle \phi(\mathbf{x}), \theta_t - \theta_0 \rangle - h(\mathbf{x}; \theta_t) + h(\mathbf{x}'; \theta_t) - \langle \phi(\mathbf{x}'), \theta_t - \theta_0 \rangle|$$

$$\leq |\langle \phi(\mathbf{x}), \theta_t - \theta_0 \rangle - h(\mathbf{x}; \theta_t)| + |h(\mathbf{x}'; \theta_t) - \langle \phi(\mathbf{x}'), \theta_t - \theta_0 \rangle| \qquad (35)$$

$$\leq 2C_2 m^{-1/6}\sqrt{\log m}L^3(\frac{t}{\lambda})^{4/3}$$

$$= 2\epsilon'_{m,t}$$

□

### E.2 PROOF OF LEMMA 12

For simplicity, we denote $\widetilde{\phi}_{0,s,i} \triangleq g(\mathbf{x}_{s,i}^1; \theta_0) - g(\mathbf{x}_{s,i}^2; \theta_0)$, $\widetilde{\phi}_{t,s,i} = g(\mathbf{x}_{s,i}^1; \theta_t) - g(\mathbf{x}_{s,i}^2; \theta_t)$ and $\widetilde{h}_{t,s,i} \triangleq h(\mathbf{x}_{s,i}^1; \theta_t) - h(\mathbf{x}_{s,i}^2; \theta_t)$

**Lemma 12.** Let $\beta_T \triangleq \frac{1}{\kappa_\mu}\sqrt{\widetilde{d} + 2\log(1/\delta)}$. Assuming that the conditions on $m$ from Eq. 34 are satisfied. With probability of at least $1 - \delta$, we have that

$$\sqrt{m}\,\|\theta_{r^*} - \theta_t\|_{V_{t-1}} \le \beta_T + B\sqrt{\frac{\lambda}{\kappa_\mu}} + 1, \forall t \in [T].$$

For any $\theta_{r'} \in \mathbb{R}^p$, define

$$G_t(\theta_{r'}) = \frac{1}{m}\sum_{s=1}^{t-1}\sum_{i=1}^{k}\left[\mu\left(\left\langle\theta_{r'} - \theta_0, \widetilde{\phi}_{0,s,i}\right\rangle\right) - \mu\left(\left\langle\theta_{r^*} - \theta_0, \widetilde{\phi}_{0,s,i}\right\rangle\right)\right]\widetilde{\phi}_{0,s,i} + \lambda(\theta_{r'} - \theta_0). \tag{36}$$

**Lemma 13.** Choose $\lambda > 0$ such that $\lambda/\kappa_\mu \ge 1$. Define $V_{t-1} \triangleq \sum_{s=1}^{t-1}\sum_{i=1}^{k}\widetilde{\phi}_{0,s,i}\widetilde{\phi}_{0,s,i}^\top\frac{1}{m} + \frac{\lambda}{\kappa_\mu}\mathbf{I}$

$$\|\theta_{r^*} - \theta_t\|_{V_{t-1}} \le \frac{1}{\kappa_\mu}\|G_t(\theta_t)\|_{V_{t-1}^{-1}} + \sqrt{\frac{\lambda}{\kappa_\mu}}\frac{B}{\sqrt{m}}$$

*Proof.* Let $\lambda' \in (0,1)$. For any $\theta_{r_1'}, \theta_{r_2'} \in \mathbb{R}^p$, setting $\theta_{\bar{r}} = \lambda'\theta_{r_1'} + (1 - \lambda')\theta_{r_2'}$ and using the mean value theorem, we get:

$$\begin{aligned}
G_t(\theta_{r_1'}) - G_t(\theta_{r_2'}) &= \left[\frac{1}{m}\sum_{s=1}^{t-1}\sum_{i=1}^{k}\dot{\mu}(\left\langle\theta_{\bar{r}} - \theta_0, \widetilde{\phi}_{0,s,i}\right\rangle)\widetilde{\phi}_{0,s,i}\widetilde{\phi}_{0,s,i}^\top + \lambda\mathbf{I}_p\right](\theta_{r_1'} - \theta_{r_2'}) \\
&\ge \kappa_\mu\left[\frac{1}{m}\sum_{s=1}^{t-1}\sum_{i=1}^{k}\widetilde{\phi}_{0,s,i}\widetilde{\phi}_{0,s,i}^\top + \frac{\lambda}{\kappa_\mu}\mathbf{I}_p\right](\theta_{r_1'} - \theta_{r_2'}) \\
&= \kappa_\mu V_{t-1}(\theta_{r_1'} - \theta_{r_2'})
\end{aligned}$$

Note that $G_t(\theta_{r^*}) = \lambda(\theta_{r^*} - \theta_0)$ and $r_t$ is the estimate of $r^*$ at the beginning of the iteration $t$ and $r_{t,s,i} = \left\langle\theta_t - \theta_0, \widetilde{\phi}_{0,s,i}\right\rangle$. Now using the equation above,

$$\begin{aligned}
\|G_t(\theta_t) - \lambda(\theta_{r^*} - \theta_0)\|_{V_{t-1}^{-1}}^2 &= \|G_t(\theta_{r^*}) - G_t(\theta_t)\|_{V_{t-1}^{-1}}^2 \\
&\ge (\kappa_\mu V_{t-1}(\theta_{r^*} - \theta_t))^\top V_{t-1}^{-1}\kappa_\mu V_{t-1}(\theta_{r^*} - \theta_t) \\
&= \kappa_\mu^2(\theta_{r^*} - \theta_t)^\top V_{t-1}V_{t-1}^{-1}V_{t-1}(\theta_{r^*} - \theta_t) \\
&= \kappa_\mu^2\|\theta_{r^*} - \theta_t\|_{V_{t-1}}^2
\end{aligned}$$

This allows us to show that

$$\|\theta_{r^*} - \theta_t\|_{V_{t-1}} \le \frac{1}{\kappa_\mu}\|G_t(\theta_t) - \lambda(\theta_{r^*} - \theta_0)\|_{V_{t-1}^{-1}} \le \frac{1}{\kappa_\mu}\|G_t(\theta_t)\|_{V_{t-1}^{-1}} + \frac{1}{\kappa_\mu}\|\lambda(\theta_{r^*} - \theta_0)\|_{V_{t-1}^{-1}} \tag{37}$$

in which we have made use of the triangle inequality.

Note that we choose $\lambda$ such that $\frac{\lambda}{\kappa_\mu} \ge 1$. This allows us to show that $V_{t-1} \succeq \frac{\lambda}{\kappa_\mu}I$ and hence $V_{t-1}^{-1} \preceq \frac{\kappa_\mu}{\lambda}I$ Recall that Lemma 7 tells us that $\|\theta_{r^*} - \theta_0\|_2 \le \frac{B}{\sqrt{m}}$, which tells us that

$$
\begin{aligned}
\frac{1}{\kappa_\mu} \left\| \lambda(\theta_{r^*} - \theta_0) \right\|_{V_{t-1}^{-1}} &= \frac{\lambda}{\kappa_\mu} \sqrt{(\theta_{r^*} - \theta_0)^\top V_{t-1}^{-1} (\theta_{r^*} - \theta_0)} \\
&= \frac{\lambda}{\kappa_\mu} \sqrt{(\theta_{r^*} - \theta_0)^\top \frac{\kappa_\mu}{\lambda} (\theta_{r^*} - \theta_0)} \\
&\leq \sqrt{\frac{\lambda}{\kappa_\mu}} \left\| \theta_{r^*} - \theta_0 \right\|_2 \\
&\leq \sqrt{\frac{\lambda}{\kappa_\mu}} \frac{B}{\sqrt{m}}
\end{aligned}
\tag{38}
$$

This completes the proof. $\square$

Recall that we denote $y_{s,i} = \mu(r^*(\mathbf{x}_{s,i}^1) - r^*(\mathbf{x}_{s,i}^2)) + \epsilon_{s,i}$ After that, we can derive an upper bound from the Lemma above.

$$
\begin{aligned}
\frac{1}{\kappa_\mu} \left\| G_t(\theta_t) \right\|_{V_{t-1}^{-1}} &= \frac{1}{\kappa_\mu} \left\| \frac{1}{m} \sum_{s=1}^{t-1} \sum_{i=1}^{k} \left[ \mu(\langle \theta_t - \theta_0, \widetilde{\phi}_{0,s,i} \rangle) - \mu(\langle \theta_{r^*} - \theta_0, \widetilde{\phi}_{0,s,i} \rangle) \right] \widetilde{\phi}_{0,s,i} + \lambda(\theta_t - \theta_0) \right\|_{V_{t-1}^{-1}} \\
&= \frac{1}{\kappa_\mu} \left\| \frac{1}{m} \sum_{s=1}^{t-1} \sum_{i=1}^{k} \left[ \mu(r_{t,s,i}) - \mu(r^*(\mathbf{x}_{s,i}^1) - r^*(\mathbf{x}_{s,i}^2)) \right] \widetilde{\phi}_{0,s,i} + \lambda(\theta_t - \theta_0) \right\|_{V_{t-1}^{-1}} \\
&= \frac{1}{\kappa_\mu} \left\| \frac{1}{m} \sum_{s=1}^{t-1} \sum_{i=1}^{k} \left[ \mu(r_{t,s,i}) - (y_{s,i} - \epsilon_{s,i}) \right] \widetilde{\phi}_{0,s,i} + \lambda(\theta_t - \theta_0) \right\|_{V_{t-1}^{-1}} \\
&= \frac{1}{\kappa_\mu} \left\| \frac{1}{m} \sum_{s=1}^{t-1} \sum_{i=1}^{k} (\mu(r_{t,s,i}) - y_{s,i}) \widetilde{\phi}_{0,s,i} + \frac{1}{m} \sum_{s=1}^{t-1} \sum_{i=1}^{k} \epsilon_{s,i} \widetilde{\phi}_{0,s,i} + \lambda(\theta_t - \theta_0) \right\|_{V_{t-1}^{-1}} \\
&\leq \frac{1}{\kappa_\mu} \left\| \frac{1}{m} \sum_{s=1}^{t-1} \sum_{i=1}^{k} (\mu(r_{t,s,i}) - y_{s,i}) \widetilde{\phi}_{0,s,i} + \lambda(\theta_t - \theta_0) \right\|_{V_{t-1}^{-1}} + \frac{1}{\kappa_\mu} \left\| \frac{1}{m} \sum_{s=1}^{t-1} \sum_{i=1}^{k} \epsilon_{s,i} \widetilde{\phi}_{0,s,i} \right\|_{V_{t-1}^{-1}}
\end{aligned}
\tag{39}
$$

Next, we derive an upper bound on the first term in 39. For simplicity, we define:

$$
A_1 \triangleq \frac{1}{m} \sum_{s=1}^{t-1} \sum_{i=1}^{k} (\mu(r_{t,s,i}) - y_{s,i})(\widetilde{\phi}_{0,s,i} - \widetilde{\phi}_{t,s,i}), \quad A_2 \triangleq \frac{1}{m} \sum_{s=1}^{t-1} \sum_{i=1}^{k} (\mu(r_{t,s,i}) - \mu(\widetilde{h}_{t,s,i})) \widetilde{\phi}_{t,s,i}
\tag{40}
$$

Now, the above equation can be decomposed as:

$$\left\| \frac{1}{m} \sum_{s=1}^{t-1} \sum_{i=1}^{k} (\mu(r_{t,s,i}) - y_{s,i}) \widetilde{\phi}_{0,s,i} + \lambda(\theta_t - \theta_0) \right\|_{V_{t-1}^{-1}}$$

$$= \left\| \frac{1}{m} \sum_{s=1}^{t-1} \sum_{i=1}^{k} (\mu(r_{t,s,i}) - y_{s,i}) (\widetilde{\phi}_{0,s,i} + \widetilde{\phi}_{t,s,i} - \widetilde{\phi}_{t,s,i}) + \lambda(\theta_t - \theta_0) \right\|_{V_{t-1}^{-1}}$$

$$= \left\| \frac{1}{m} \sum_{s=1}^{t-1} \sum_{i=1}^{k} (\mu(r_{t,s,i}) - y_{s,i}) \widetilde{\phi}_{t,s,i} + \lambda(\theta_t - \theta_0) + A_1 \right\|_{V_{t-1}^{-1}}$$

$$= \left\| \frac{1}{m} \sum_{s=1}^{t-1} \sum_{i=1}^{k} (\mu(r_{t,s,i}) + \mu(\widetilde{h}_{t,s,i}) - \mu(\widetilde{h}_{t,s,i}) - y_{s,i}) \widetilde{\phi}_{t,s,i} + \lambda(\theta_t - \theta_0) + A_1 \right\|_{V_{t-1}^{-1}} \quad (41)$$

$$= \left\| \frac{1}{m} \sum_{s=1}^{t-1} \sum_{i=1}^{k} (\mu(\widetilde{h}_{t,s,i}) - y_{s,i}) \widetilde{\phi}_{t,s,i} + \lambda(\theta_t - \theta_0) + A_2 + A_1 \right\|_{V_{t-1}^{-1}}$$

$$\overset{(a)}{=} \|A_2 + A_1\|_{V_{t-1}^{-1}}$$

$$\leq \|A_2\|_{V_{t-1}^{-1}} + \|A_1\|_{V_{t-1}^{-1}}$$

$$\leq \sqrt{\frac{\kappa_\mu}{\lambda}} \|A_2\|_2 + \sqrt{\frac{\kappa_\mu}{\lambda}} \|A_1\|_2$$

Note that we have step $(a)$ because:

$$\frac{1}{m} \sum_{s=1}^{t-1} \sum_{i=1}^{k} \left( \mu(\widetilde{h}_{t,s,i}) - y_{s,i} \right) \widetilde{\phi}_{t,s,i} + \lambda(\theta_t - \theta_0)$$

$$= \frac{1}{m} \sum_{s=1}^{t-1} \sum_{i=1}^{k} \left( \mu(h(\mathbf{x}_{s,i}^1; \theta_t) - h(\mathbf{x}_{s,i}^2; \theta_t)) - y_{s,i} \right) (g(\mathbf{x}_{s,i}^1; \theta_t) - g(\mathbf{x}_{s,i}^2; \theta_t)) + \lambda(\theta_t - \theta_0) \quad (42)$$

$$= 0$$

which is ensured by the loss function and the way we train our NN. Next, we derive an upper bound on the norm of $A_1$. To begin with, we have that

$$\left\| \widetilde{\phi}_{0,s,i} - \widetilde{\phi}_{t,s,i} \right\|_2 = \left\| g(\mathbf{x}_{s,i}^1; \theta_0) - g(\mathbf{x}_{s,i}^2; \theta_0) - g(\mathbf{x}_{s,i}^1; \theta_t) + g(\mathbf{x}_{s,i}^2; \theta_t) \right\|_2$$

$$\leq \left\| g(\mathbf{x}_{s,i}^1; \theta_0) - g(\mathbf{x}_{s,i}^1; \theta_t) \right\|_2 + \left\| g(\mathbf{x}_{s,i}^2; \theta_0) - g(\mathbf{x}_{s,i}^2; \theta_t) \right\|_2$$

$$\leq 2C_1 m^{1/3} \sqrt{\log m} \left( \frac{Ct}{\lambda} \right)^{1/3} L^{7/2}$$

in which the last inequality follows from Lemma 9. Now the norm of $A_1$ can be bounded as:

$$
\begin{aligned}
\|A_1\|_2 &= \left\| \frac{1}{m} \sum_{s=1}^{t-1} \sum_{i=1}^{k} (\mu(r_{t,s,i}) - y_{s,i})(\widetilde{\phi}_{0,s,i} - \widetilde{\phi}_{t,s,i}) \right\|_2 \\
&\leq \frac{1}{m} \sum_{s=1}^{t-1} \sum_{i=1}^{k} \left\| (\mu(r_{t,s,i}) - y_{s,i})(\widetilde{\phi}_{0,s,i} - \widetilde{\phi}_{t,s,i}) \right\|_2 \\
&= \frac{1}{m} \sum_{s=1}^{t-1} \sum_{i=1}^{k} |\mu(r_{t,s,i}) - y_{s,i}| \left\| \widetilde{\phi}_{0,s,i} - \widetilde{\phi}_{t,s,i} \right\|_2 \qquad (43) \\
&\leq \frac{1}{m} \sum_{s=1}^{t-1} \sum_{i=1}^{k} \left\| \widetilde{\phi}_{0,s,i} - \widetilde{\phi}_{t,s,i} \right\|_2 \\
&\leq \frac{1}{m} \sum_{s=1}^{t-1} \sum_{i=1}^{k} 2C_1 m^{1/3} \sqrt{\log m} \left( \frac{Ct}{\lambda} \right)^{1/3} L^{7/2} \\
&= m^{-2/3} \sqrt{\log m} t^{4/3} 2C_1 k \lambda^{-1/3} L^{7/2}
\end{aligned}
$$

Next, we proceed to bound the norm of $A_2$. Let $\lambda' \in (0,1)$, and let $a_{t,s,i} = \lambda' r_{t,s,i} + (1 - \lambda')\widetilde{h}_{t,s,i}$. Following the mean-value theorem, we have for some $\lambda'$ that

$$
\mu(r_{t,s,i}) - \mu(\widetilde{h}_{t,s,i}) = (r_{t,s,i} - \widetilde{h}_{t,s,i})\dot{\mu}(a_{t,s,i})
$$

Note that $\dot{\mu}(a_{t,s,i}) \leq L_\mu$ which follows our Assumption. This allows us to show that

$$
\begin{aligned}
|\mu(r_{t,s,i}) - \mu(\widetilde{h}_{t,s,i})| &= \left| (r_{t,s,i} - \widetilde{h}_{t,s,i})\dot{\mu}(a_{t,s,i}) \right| \\
&= |r_{t,s,i} - \widetilde{h}_{t,s,i}||\dot{\mu}(a_{t,s,i})| \\
&\leq L_\mu |r_{t,s,i} - \widetilde{h}_{t,s,i}| \\
&= L_\mu \left| \langle \theta_t - \theta_0, g(\mathbf{x}_{s,i}^1; \theta_0) \rangle - \langle \theta_t - \theta_0, g(\mathbf{x}_{s,i}^2; \theta_0) \rangle - (h(\mathbf{x}_{s,i}^1; \theta_t) - h(\mathbf{x}_{s,i}^2; \theta_t)) \right| \\
&= L_\mu \left( \left| \langle \theta_t - \theta_0, g(\mathbf{x}_{s,i}^1; \theta_0) \rangle - h(\mathbf{x}_{s,i}^1; \theta_t) \right| + \left| h(\mathbf{x}_{s,i}^2; \theta_t) - \langle \theta_t - \theta_0, g(\mathbf{x}_{s,i}^2; \theta_0) \rangle \right| \right) \\
&\overset{(a)}{\leq} 2L_\mu C_2 m^{-1/6} \sqrt{\log m} L^3 \left( \frac{t}{\lambda} \right)^{4/3}
\end{aligned}
$$

$$(44)$$

where step $(a)$ follows from Lemma 10 . We can also notice the fact that $\left\| \widetilde{\phi}_{t,s,i} \right\|_2 = \left\| g(\mathbf{x}_{s,i}^1; \theta_t) - g(\mathbf{x}_{s,i}^2; \theta_t) \right\|_2 \leq \left\| g(\mathbf{x}_{s,i}^1; \theta_t) \right\|_2 + \left\| g(\mathbf{x}_{s,i}^2; \theta_t) \right\|_2 \leq 2C_3\sqrt{mL}$ Then we can derive an upper bound of $A_2$ using the fact above.

$$
\begin{aligned}
\|A_2\|_2 &= \left\| \frac{1}{m} \sum_{s=1}^{t-1} \sum_{i=1}^{k} (\mu(r_{t,s,i}) - \mu(\widetilde{h}_{t,s,i}))\widetilde{\phi}_{t,s,i} \right\|_2 \\
&\leq \frac{1}{m} \sum_{s=1}^{t-1} \sum_{i=1}^{k} \left\| (\mu(r_{t,s,i}) - \mu(\widetilde{h}_{t,s,i}))\widetilde{\phi}_{t,s,i} \right\|_2 \\
&= \frac{1}{m} \sum_{s=1}^{t-1} \sum_{i=1}^{k} |\mu(r_{t,s,i}) - \mu(\widetilde{h}_{t,s,i})| \left\| \widetilde{\phi}_{t,s,i} \right\|_2 \qquad (45) \\
&\leq \frac{1}{m} \sum_{s=1}^{t-1} \sum_{i=1}^{k} 2L_\mu C_2 m^{-1/6} \sqrt{\log m} L^3 \left( \frac{t}{\lambda} \right)^{4/3} \times 2C_3\sqrt{mL} \\
&\leq 4k L_\mu C_2 C_3 m^{-2/3} \sqrt{\log m} t^{7/3} L^{7/2} \lambda^{-4/3}
\end{aligned}
$$

Lastly, we can derive an upper bound of the first term of Eq. equation 39 by combining Eq. (43), Eq. (45) and Eq. (41):

$$\frac{1}{\kappa_\mu} \left\| \frac{1}{m} \sum_{s=1}^{t-1} \sum_{i=1}^{k} (\mu(r_{t,s,i}) - y_{s,i}) \widetilde{\phi}_{0,s,i} + \lambda(\theta_t - \theta_0) \right\|_{V_{t-1}^{-1}}$$

$$= \|A_1 + A_2\|_{V_{t-1}^{-1}}$$

$$\leq \sqrt{\frac{\kappa_\mu}{\lambda}} \|A_1\|_2 + \sqrt{\frac{\kappa_\mu}{\lambda}} \|A_2\|_2$$

$$\leq \frac{1}{\sqrt{\kappa_\mu \lambda}} m^{-2/3} \sqrt{\log m} t^{4/3} 2C_1 k \lambda^{-1/3} L^{7/2} + \frac{1}{\sqrt{\kappa_\mu \lambda}} 4k L_\mu C_2 C_3 m^{-2/3} \sqrt{\log m} t^{7/3} L^{7/2} \lambda^{-4/3}$$

$$(46)$$

And then, plugging equation Eq. (46) into equation Eq. equation 39 and plugging the results from Lemma 13 we can get that

$$\|\theta_{r^*} - \theta_t\|_{V_{t-1}}$$

$$\leq \frac{1}{\kappa_\mu \sqrt{m}} \left\| \frac{1}{\sqrt{m}} \sum_{s=1}^{t-1} \sum_{i=1}^{k} \epsilon_{s,i} \widetilde{\phi}_{0,s,i} \right\|_{V_{t-1}^{-1}} + \sqrt{\frac{\lambda}{\kappa_\mu}} \frac{B}{\sqrt{m}} + \frac{1}{\sqrt{\kappa_\mu \lambda}} m^{-2/3} \sqrt{\log m} t^{4/3} 2C_1 k \lambda^{-1/3} L^{7/2}$$

$$+ \frac{1}{\sqrt{\kappa_\mu \lambda}} 4k L_\mu C_2 C_3 m^{-2/3} \sqrt{\log m} t^{7/3} L^{7/2} \lambda^{-4/3}$$

Here we define

$$\epsilon_{m,t} \triangleq B\sqrt{\frac{\lambda}{\kappa_\mu}} + \frac{1}{\sqrt{\kappa_\mu \lambda}} m^{-1/6} \sqrt{\log m} t^{4/3} 2C_1 k \lambda^{-1/3} L^{7/2} +$$

$$\frac{1}{\sqrt{\kappa_\mu \lambda}} 4k L_\mu C_2 C_3 m^{-1/6} \sqrt{\log m} t^{7/3} L^{7/2} \lambda^{-4/3}$$

$$(47)$$

It is easy to verify that as long as the condition on $m$ from are satisfied, we have that $\epsilon_{m,t} \leq B\sqrt{\frac{\lambda}{\kappa_\mu}} + 1$

This allows us to show that

$$\sqrt{m} \|\theta_{r^*} - \theta_t\| \leq \frac{1}{\kappa_\mu} \left\| \frac{1}{\sqrt{m}} \sum_{s=1}^{t-1} \sum_{i=1}^{k} \epsilon_{s,i} \widetilde{\phi}_{0,s,i} \right\|_{V_{t-1}^{-1}} + \epsilon_{m,t}$$

$$\leq \frac{1}{\kappa_\mu} \left\| \frac{1}{\sqrt{m}} \sum_{s=1}^{t-1} \sum_{i=1}^{k} \epsilon_{s,i} \widetilde{\phi}_{0,s,i} \right\|_{V_{t-1}^{-1}} + B\sqrt{\frac{\lambda}{\kappa_\mu}} + 1$$

$$(48)$$

Finally, in the next lemma, we derive an upper bound on the first term in Eq. (48)

**Lemma 14.** *Let* $\beta_T \triangleq \frac{1}{\kappa_\mu} \sqrt{\widetilde{d} + 2\log(1/\delta)}$ *With probability of at least* $1 - \delta$, *we have that*

$$\frac{1}{\kappa_\mu} \left\| \frac{1}{\sqrt{m}} \sum_{s=1}^{t-1} \sum_{i=1}^{k} \epsilon_{s,i} \widetilde{\phi}_{0,s,i} \right\|_{V_{t-1}^{-1}} \leq \beta_T$$

*Proof.* Recall that we have $V_t = \sum_{\tau=1}^{t} \sum_{i=1}^{k} \widetilde{\phi}_{0,\tau,i} \widetilde{\phi}_{0,\tau,i}^\top \frac{1}{m} + \frac{\lambda}{\kappa_\mu} \mathbf{I}$. Here we use $C_2^K$ to denote all possible pairwise combinations of the indices of $K$ arms. We denote $z_j^i(s) \triangleq \phi(\mathbf{x}_{s,i}) - \phi(\mathbf{x}_{s,j})$. Also recall we have defined that $\mathbf{H}' \triangleq \sum_{s=1}^{T} \sum_{(i,j) \in C_2^K} z_j^i(s) z_j^i(s)^\top \frac{1}{m}$. Now the determinant of $V_t$ can be upper bounded as

$$det(V_t) = det\left(\sum_{\tau=1}^{t}\sum_{i=1}^{k}\widetilde{\phi}_{0,\tau,i}\widetilde{\phi}_{0,\tau,i}^{\top}\frac{1}{m} + \frac{\lambda}{\kappa_\mu}\mathbf{I}\right)$$

$$\leq det\left(\sum_{\tau=1}^{T}\sum_{i=1}^{k}(\phi(\mathbf{x}_{\tau,i}^1) - \phi(\mathbf{x}_{\tau,i}^2))(\phi(\mathbf{x}_{\tau,i}^1) - \phi(\mathbf{x}_{\tau,i}^2))^{\top}\frac{1}{m} + \frac{\lambda}{\kappa_\mu}\mathbf{I}\right)$$

$$\leq det\left(\sum_{s=1}^{T}\sum_{(i,j)\in C_2^K} z_j^i(s)z_j^i(s)^{\top}\frac{1}{m} + \frac{\lambda}{\kappa_\mu}\mathbf{I}\right) \tag{49}$$

$$= det\left(\mathbf{H}' + \frac{\lambda}{\kappa_\mu}\mathbf{I}\right)$$

Recall that in our algorithm $V_0 = \frac{\lambda}{\kappa_\mu}\mathbf{I}$. This leads to

$$\log\frac{\det V_t}{\det V_0} \leq \log\frac{\det\left(\mathbf{H}' + \frac{\lambda}{\kappa_\mu}\mathbf{I}\right)}{\det V_0}$$

$$= \log\frac{(\lambda/\kappa_\mu)^p \det\left(\frac{\kappa_\mu}{\lambda}\mathbf{H}' + \mathbf{I}\right)}{(\lambda/\kappa_\mu)^p} \tag{50}$$

$$= \log\det\left(\frac{\kappa_\mu}{\lambda}\mathbf{H}' + \mathbf{I}\right)$$

We use $\epsilon_{s,i}$ to denote the observation noise in iteration $s \in [T] : y_{s,i} = \mu(r^*(\mathbf{x}_{s,i}^1) - r^*(\mathbf{x}_{s,i}^2)) + \epsilon_{s,i}$. Let $\mathcal{F}_{t-1}$ denote the sigma algebra generated by history $\{(\mathbf{x}_{s,i}^1, \mathbf{x}_{s,i}^2, \epsilon_{s,i})_{s\in[t-1]}, (\mathbf{x}_{t,i}^1, \mathbf{x}_{t,i}^2)_{i=1...k}\}$. Here we justify that the sequence of noise $\{\epsilon_{s,i}\}_{s=1,...,T,i=1,...,k}$ is conditionally 1-sub-Gaussian conditioned on $\mathcal{F}_{t-1}$. Note that the observation $y_{t,i}$ is equal to 1 if $\mathbf{x}_{t,i}^1$ is preferred over $\mathbf{x}_{t,i}^2$ and 0 otherwise. Therefore, the noise $\epsilon_t$ can be expressed as

$$\epsilon_{s,i} = \begin{cases} 1 - \mu(r^*(\mathbf{x}_{s,i}^1) - r^*(\mathbf{x}_{s,i}^2)), & w.p. \quad \mu(r^*(\mathbf{x}_{s,i}^1) - r^*(\mathbf{x}_{s,i}^2)) \\ -\mu(r^*(\mathbf{x}_{s,i}^1) - r^*(\mathbf{x}_{s,i}^2)), & w.p. \quad 1 - \mu(r^*(\mathbf{x}_{s,i}^1) - r^*(\mathbf{x}_{s,i}^2)) \end{cases} \tag{51}$$

It can be seen that $\epsilon_{s,i}$ is $\mathcal{F}_t$- measurable. Next, it can beverified that $\mathbb{E}[\epsilon_{s,i}|\mathcal{F}_{t-1}] = 0$. We can also note that $|\epsilon_{s,i}| \leq 1$. Therefore, we can infer that $\epsilon_{s,i}$ is conditionally 1-sub-Gaussian,

$$\mathbb{E}[exp(\lambda\epsilon_{s,i})|\mathcal{F}_t] \leq exp(\frac{\lambda^2\sigma^2}{2}), \forall\lambda\in\mathbb{R} \tag{52}$$

with $\sigma = 1$.

Next, making use of the 1-sub-sub-Guassianity of the sequence of noise $\{\epsilon_{s,i}\}$ and Theorem 1 from (Abbasi-Yadkori et al., 2011), we can show that with probability at least $1 - \delta$

$$\frac{1}{\kappa_\mu}\left\|\frac{1}{\sqrt{m}}\sum_{s=1}^{t-1}\sum_{i=1}^{k}\epsilon_{s,i}\widetilde{\phi}_{0,s,i}\right\|_{V_{t-1}^{-1}} \leq \sqrt{\log\left(\frac{\det V_{t-1}}{\det V_0}\right) + 2\log(1/\delta)}$$

$$\leq \sqrt{\log\det\left(\frac{\kappa_\mu}{\lambda}\mathbf{H}' + \mathbf{I}\right) + 2\log(1/\delta)}$$

$$\overset{(a)}{\leq} \sqrt{\widetilde{d} + 2\log(1/\delta)}$$

in which we use the definition of $\widetilde{d} = \log\det(\mathbf{I} + \frac{\kappa_\mu}{\lambda}\mathbf{H}')$ in step $(a)$. This completes the proof.

$\square$

Finally, plugging Lemma 14 into 48, we complete the proof of Lemma 12:

$$\sqrt{m}\|\theta_r - \theta_t\|_{V_{t-1}} \leq \beta_T + B\sqrt{\frac{\lambda}{\kappa_\mu} + 1}, \forall t\in[T].$$

### E.3 PROOF OF THEOREM 2

**Definition 3.** *For simplicity, we define* $\nu_T = (\beta_T + B\sqrt{\frac{\lambda}{\kappa_\nu}} + 1)\frac{\kappa_\nu}{\lambda}$ *and*
$$\sigma_t(S_t^1, S_t^2) \triangleq \frac{\lambda}{\kappa_\mu} \sum_{i=1}^{k} \left\| \frac{1}{\sqrt{m}}(\phi(\mathbf{x}_{t,i}^1) - \phi(\mathbf{x}_{t,i}^2)) \right\|_{V_{t-1}^{-1}}.$$

**Lemma 15.** *Let* $\delta \in (0, 1)$, $\epsilon_{m,t}' = C_2 m^{-1/6}\sqrt{\log m} L^3(\frac{t}{\lambda})^{4/3}$ *for some absolute constant* $C_2 > 0$. *As long as* $m \geq poly(T, L, K, 1/\kappa_\mu, L_\mu, 1/\lambda_0, 1/\lambda, \log(1/\delta))$, *then with probability of at least* $1 - \delta$,
$$|[r^*(\mathbf{x}) - r^*(\mathbf{x}')] - [h(\mathbf{x}; \theta_t) - h(\mathbf{x}'; \theta_t)]| \leq \sigma_{t-1}(\mathbf{x}, \mathbf{x}') + 2\epsilon_{m,t}'$$

*for all* $\mathbf{x}, \mathbf{x}' \in X_t$, $t \in [T]$.

*Proof.* Denote $\phi(\mathbf{x}) = g(\mathbf{x}; \theta_0)$, recall that $r^*(\mathbf{x}) = \langle g(\mathbf{x}; \theta_0), \theta_{r^*} - \theta_0 \rangle = \langle \phi(\mathbf{x}), \theta_{r^*} - \theta_0 \rangle$ for all $\mathbf{x} \in \mathcal{X}_t, t \in [T]$. For $\mathbf{x}, \mathbf{x}' \in \mathcal{X}_t, t \in [T]$ we have that

$$
\begin{aligned}
&|r^*(\mathbf{x}) - r^*(\mathbf{x}') - \langle \phi(\mathbf{x}) - \phi(\mathbf{x}'), \theta_t - \theta_0 \rangle| \\
&= |\langle \phi(\mathbf{x}) - \phi(\mathbf{x}'), \theta_{r^*} - \theta_0 \rangle - \langle \phi(\mathbf{x}) - \phi(\mathbf{x}'), \theta_t - \theta_0 \rangle| \\
&= |\langle \phi(\mathbf{x}) - \phi(\mathbf{x}'), \theta_{r^*} - \theta_t \rangle| \\
&= \left| \left\langle \frac{1}{\sqrt{m}}(\phi(\mathbf{x}) - \phi(\mathbf{x}')), \sqrt{m}(\theta_{r^*} - \theta_0) \right\rangle \right| \\
&\leq \left\| \frac{1}{\sqrt{m}}(\phi(\mathbf{x}) - \phi(\mathbf{x}')) \right\|_{V_{t-1}^{-1}} \sqrt{m} \|\theta_{r^*} - \theta_t\|_{V_{t-1}} \\
&\overset{(a)}{\leq} \left\| \frac{1}{\sqrt{m}}(\phi(\mathbf{x}) - \phi(\mathbf{x}')) \right\|_{V_{t-1}^{-1}} \left( \beta_T + B\sqrt{\frac{\lambda}{\kappa_\mu}} + 1 \right)
\end{aligned}
\tag{53}
$$

in which we use Lemma 12 in step (a). Now making use of the equation above and Lemma 10, we have that

$$
\begin{aligned}
&|r^*(\mathbf{x}) - r^*(\mathbf{x}') - (h(\mathbf{x}; \theta_t) - h(\mathbf{x}'; \theta_t))| \\
&= |r^*(\mathbf{x}) - r^*(\mathbf{x}') - \langle \phi(\mathbf{x}) - \phi(\mathbf{x}'), \theta_t - \theta_0 \rangle \\
&\quad + \langle \phi(\mathbf{x}) - \phi(\mathbf{x}'), \theta_t - \theta_0 \rangle - (h(\mathbf{x}; \theta_t) - h(\mathbf{x}'; \theta_t))| \\
&\leq |r^*(\mathbf{x}) - r^*(\mathbf{x}') - \langle \phi(\mathbf{x}) - \phi(\mathbf{x}'), \theta_t - \theta_0 \rangle| \\
&\quad + |\langle \phi(\mathbf{x}) - \phi(\mathbf{x}'), \theta_t - \theta_0 \rangle - (h(\mathbf{x}; \theta_t) - h(\mathbf{x}'; \theta_t))| \\
&\leq \left\| \frac{1}{\sqrt{m}}(\phi(\mathbf{x}) - \phi(\mathbf{x}')) \right\|_{V_{t-1}^{-1}} \left( \beta_T + B\sqrt{\frac{\lambda}{\kappa_\mu}} + 1 \right) + 2\epsilon_{m,t}'
\end{aligned}
\tag{54}
$$

This completes the proof of Theorem. $\qquad\square$

**Lemma 2.** *Let* $\delta \in (0, 1)$, $\epsilon_{m,t}' = C_2 m^{-1/6}\sqrt{\log m} L^3(\frac{t}{\lambda})^{4/3}$ *for some absolute constant* $C_2 > 0$. *As long as* $m \geq poly(T, L, K, 1/\kappa_\mu, L_\mu, 1/\lambda_0, 1/\lambda, \log(1/\delta))$, *Let the context of two super arm* $S_t^1$ *and* $S_t^2$ *to be* $\mathcal{X}_t(S_t^1) = \{\mathbf{x}_{t,i}^1\}_{i=1}^k$, $\mathcal{X}_t(S_t^2) = \{\mathbf{x}_{t,i}^2\}_{i=1}^k$, *then with probability of at least* $1 - \delta$,
$$\left| (f(S_t^1, \mathbf{r}^*) - f(S_t^2, \mathbf{r}^*)) - (f(S_t^1, \mathbf{r}_t) - f(S_t^2, \mathbf{r}_t)) \right| \leq \nu_T \sigma_{t-1}(S_t^1, S_t^2) + 2k\epsilon_{m,t}'$$

*for all* $t \in [T]$

*Proof.*

$$\left| \left( f(S_t^1, \mathbf{r}^*) - f(S_t^2, \mathbf{r}^*) \right) - \left( f(S_t^1, \mathbf{r}_t) - f(S_t^2, \mathbf{r}_t) \right) \right|$$

$$= \left| \left[ \sum_{i=1}^k r^*(\mathbf{x}_{t,i}^1) - \sum_{i=1}^k r^*(\mathbf{x}_{t,i}^2) - \sum_{i=1}^k r_t(\mathbf{x}_{t,i}^1) + \sum_{i=1}^k r_t(\mathbf{x}_{t,i}^2) \right] \right|$$

$$= \left| \sum_{i=1}^k \left[ \left( r^*(\mathbf{x}_{t,i}^1) - r^*(\mathbf{x}_{t,i}^2) \right) - \left( r_t(\mathbf{x}_{t,i}^1) - r_t(\mathbf{x}_{t,i}^2) \right) \right] \right| \tag{55}$$

$$\leq \sum_{i=1}^k \left| \left[ \left( r^*(\mathbf{x}_{t,i}^1) - r^*(\mathbf{x}_{t,i}^2) \right) - \left( h_t(\mathbf{x}_{t,i}^1; \theta_t) - h_t(\mathbf{x}_{t,i}^2; \theta_t) \right) \right] \right|$$

$$\leq \sum_{i=1}^k \left( \nu_T \sigma_{t-1}(\mathbf{x}_{t,i}^1, \mathbf{x}_{t,i}^2) + 2\epsilon_{m,t}' \right)$$

$$= \nu_T \sigma_{t-1}(S_t^1, S_t^2) + 2k\epsilon_{m,t}'.$$

$\square$

Now we can analyze the regret. To begin with, we have

$$reg_t = 2\mathrm{opt}_{\mathbf{r}^*} - \left( f(S_t^1, \mathbf{r}^*) + f(S_t^2, \mathbf{r}^*) \right)$$

$$= f(S^*, \mathbf{r}^*) - f(S_t^1, \mathbf{r}^*) + f(S^*, \mathbf{r}^*) - f(S_t^2, \mathbf{r}^*)$$

$$\overset{(a)}{\leq} f(S^*, \mathbf{r}_t) - f(S_t^1, \mathbf{r}_t) + \nu_T \sigma_{t-1}(S^*, S_t^1) + 2k\epsilon_{m,t}' + f(S^*, \mathbf{r}_t) - f(S_t^2, \mathbf{r}_t) + \nu_T \sigma_{t-1}(S^*, S_t^2) + 2k\epsilon_{m,t}'$$

$$\overset{(b)}{\leq} f(S^*, \mathbf{r}_t) - f(S_t^1, \mathbf{r}_t) + \nu_T \sigma_{t-1}(S^*, S_t^1) + 4k\epsilon_{m,t}' +$$

$$\quad f(S^*, \mathbf{r}_t) - f(S_t^1, \mathbf{r}_t) + f(S_t^1, \mathbf{r}_t) - f(S_t^2, \mathbf{r}_t) + \nu_T \sigma_{t-1}(S^*, S_t^1) + \nu_T \sigma_{t-1}(S_t^1, S_t^2)$$

$$= 2 \left[ f(S^*, \mathbf{r}_t) - f(S_t^1, \mathbf{r}_t) + \nu_T \sigma_{t-1}(S^*, S_t^1) \right] + f(S_t^1, \mathbf{r}_t) - f(S_t^2, \mathbf{r}_t) + \nu_T \sigma_{t-1}(S_t^1, S_t^2) + 4k\epsilon_{m,t}'$$

$$\overset{(c)}{\leq} 2 \left[ f(S_t^2, \mathbf{r}_t) - f(S_t^1, \mathbf{r}_t) + \nu_T \sigma_{t-1}(S_t^2, S_t^1) \right] + f(S_t^1, \mathbf{r}_t) - f(S_t^2, \mathbf{r}_t) + \nu_T \sigma_{t-1}(S_t^1, S_t^2) + 4k\epsilon_{m,t}'$$

$$= f(S_t^2, \mathbf{r}_t) - f(S_t^1, \mathbf{r}_t) + 3\nu_T \sigma_{t-1}(S_t^1, S_t^2) + 4k\epsilon_{m,t}'$$

$$\overset{(d)}{\leq} 3\nu_T \sigma_{t-1}(S_t^1, S_t^2) + 4k\epsilon_{m,t}'$$

$$= 3(\beta_T + B\sqrt{\frac{\lambda}{\kappa_\nu}} + 1) \sum_{i=1}^k \left\| \frac{1}{\sqrt{m}} (\phi(\mathbf{x}_{t,i}^1) - \phi(\mathbf{x}_{t,i}^2)) \right\|_{V_{t-1}^{-1}} + 4k\epsilon_{m,t}'$$

$$\tag{56}$$

Step (a) follows from Eq. (55), step (b) follows from the fact that $\sigma_{t-1}(S_t^*, S_t^2) \leq \sigma_{t-1}(S_t^*, S_t^1) + \sigma_{t-1}(S_t^1, S_t^2)$ using trangle inequality, step (c) follows from the way in which $S_t^2$ is selected:

$$S_t^2 = \arg\max_{S \in \mathcal{S}} f(S, \mathbf{r}_t) + \nu_T \sigma_{t-1}(S, S_t^1).$$

step (d) follows from the way in which $S_t^1$ is selected:

$$S_t^1 = \arg\max_{S \in \mathcal{S}} f(S, \mathbf{r}_t).$$

Recall that $V_t = \sum_{\tau=1}^t \sum_{i=1}^k \widetilde{\phi}(x_{\tau,i}^1)\widetilde{\phi}(x_{\tau,i}^2)^\top \frac{1}{m} + \frac{\lambda}{\kappa_\mu}\mathbf{I}$. It is easy to verify that $V_{t-1} \succeq \frac{\lambda}{\kappa_\mu}I$ and hence $V_{t-1}^{-1} \preceq \frac{\lambda}{\kappa_\mu}I$. Therefore, for $\forall x_{t,i}^1, x_{t,i}^2 \in \mathbf{X}_t$, it is easy to verify that

$$
\begin{aligned}
\frac{\lambda}{\kappa_\mu} \left\| \frac{1}{\sqrt{m}}(\phi(x_{t,i}^1) - \phi(x_{t,i}^2)) \right\|_{V_{t-1}^{-1}}^2 &= \frac{\lambda}{\kappa_\mu} \frac{1}{m}(\phi(x_{t,i}^1) - \phi(x_{t,i}^2))^\top V_{t-1}^{-1}(\phi(x_{t,i}^1) - \phi(x_{t,i}^2)) \\
&\leq \frac{\lambda}{\kappa_\mu} \frac{1}{m} \frac{\kappa_\mu}{\lambda}(\phi(x_{t,i}^1) - \phi(x_{t,i}^2))^\top (\phi(x_{t,i}^1) - \phi(x_{t,i}^2)) \\
&= \frac{1}{m}(\phi(x_{t,i}^1) - \phi(x_{t,i}^2))^\top (\phi(x_{t,i}^1) - \phi(x_{t,i}^2)) \\
&= \frac{1}{m} \left\| \phi(x_{t,i}^1) - \phi(x_{t,i}^2) \right\|_2^2 \\
&\leq c_0
\end{aligned}
$$

in which we have denoted $c_0 > 0$ as an absolute constant such that $\frac{1}{m} \left\| \phi(x_{t,i}^1) - \phi(x_{t,i}^2) \right\|_2^2 \leq c_0, x_{t,i}^1, x_{t,i}^2 \in \mathcal{X}_t, t \in [T]$. Note that this is similar to the standard assumption in the literature that the value of the NTK is upper bounded by a constant (Kassraie & Krause, 2022). This implies that $\frac{\lambda}{\kappa_\mu} \left\| \frac{1}{\sqrt{m}}(\phi(x_{t,i}^1) - \phi(x_{t,i}^2)) \right\|_{V_{t-1}^{-1}}^2 / c_0 \leq 1$ for some constant $c_0 \geq 1$. Recall that we choose $\lambda$ such that $\lambda/\kappa_\mu \geq 1$. Note that for any $\alpha \in [0,1]$, we have that $\alpha/2 \leq \log(1+\alpha)$. Then we have that

$$
\begin{aligned}
\frac{1}{2} \frac{\left\| \frac{1}{\sqrt{m}}(\phi(x_{t,i}^1) - \phi(x_{t,i}^2)) \right\|_{V_{t-1}^{-1}}^2}{c_0} &\leq \log\left(1 + \frac{1}{c_0} \left\| \frac{1}{\sqrt{m}}(\phi(x_{t,i}^1) - \phi(x_{t,i}^2)) \right\|_{V_{t-1}^{-1}}^2\right) \\
&\leq \log\left(1 + \left\| \frac{1}{\sqrt{m}}(\phi(x_{t,i}^1) - \phi(x_{t,i}^2)) \right\|_{V_{t-1}^{-1}}^2\right)
\end{aligned}
$$

which leads to

$$
\left\| \frac{1}{\sqrt{m}}(\phi(x_{t,i}^1) - \phi(x_{t,i}^2)) \right\|_{V_{t-1}^{-1}}^2 \leq 2c_0 \log\left(1 + \frac{\kappa_\mu}{\lambda}\frac{\lambda}{\kappa_\mu} \left\| \frac{1}{\sqrt{m}}(\phi(x_{t,i}^1) - \phi(x_{t,i}^2)) \right\|_{V_{t-1}^{-1}}^2\right) \quad (57)
$$

Then we can show that

$$
\sum_{s=1}^t \sum_{i=1}^k \log\left(1 + \frac{\kappa_\mu}{\lambda}\frac{\lambda}{\kappa_\mu} \left\| \frac{1}{\sqrt{m}}(\phi(x_{s,i}^1) - \phi(x_{s,i}^2)) \right\|_{V_{t-1}^{-1}}^2\right) = \log\det\left(\mathbf{I} + \frac{\kappa_\mu}{\lambda}\mathbf{K}_t\right)
$$

in which $\mathbf{K}_t$ is a $kt \times kt$ matrix. For $\forall i \in [kt]$, define $t_i = \lfloor \frac{i}{k} \rfloor$ and $\bar{i} = i - t_i$. And $\mathbf{K}_t[i,j] = \frac{1}{m}(\phi(x_{t_i,\bar{i}}^1) - \phi(x_{t_i,\bar{i}}^2))^\top (\phi(x_{t_j,\bar{j}}^1) - \phi(x_{t_j,\bar{j}}^2))$. Define the $p \times kt$ matrix $\mathbf{J}_t = [\frac{1}{\sqrt{m}}(\phi(x_{\tau,i}^1) - \phi(x_{\tau,i}^2))]_{\tau=1,\dots t, \ i=1,\dots k}$ And we have $\mathbf{K}_t = \mathbf{J}_t^\top \mathbf{J}_t$ This allows us to show that

$$
\sum_{s=1}^t \sum_{i=1}^k \log\left(1 + \frac{\kappa_\mu}{\lambda}\frac{\lambda}{\kappa_\mu} \left\| \frac{1}{\sqrt{m}}(\phi(x_{s,i}^1) - \phi(x_{s,i}^2)) \right\|_{V_{t-1}^{-1}}^2\right) = \log\det\left(\mathbf{I} + \frac{\kappa_\mu}{\lambda}\mathbf{K}_t\right)
$$

in which $\mathbf{K}_t$ is a $kt \times kt$ matrix. For $\forall i \in [kt]$, define $t_i = \lfloor \frac{i}{k} \rfloor$ and $\bar{i} = i - t_i$. And $\mathbf{K}_t[i,j] = \frac{1}{m}(\phi(x_{t_i,\bar{i}}^1) - \phi(x_{t_i,\bar{i}}^2))^\top (\phi(x_{t_j,\bar{j}}^1) - \phi(x_{t_j,\bar{j}}^2))$. Define the $p \times kt$ matrix $\mathbf{J}_t = [\frac{1}{\sqrt{m}}(\phi(x_{\tau,i}^1) - \phi(x_{\tau,i}^2))]_{\tau=1,\dots t, \ i=1,\dots k}$ And we have $\mathbf{K}_t = \mathbf{J}_t^\top \mathbf{J}_t$ This allows us to show that

$$\sum_{s=1}^{t}\sum_{i=1}^{k}\log\left(1+\frac{\kappa_\mu}{\lambda}\left(\frac{\lambda}{\kappa_\mu}\left\|\frac{1}{\sqrt{m}}(\phi(x_{s,i}^1)-\phi(x_{s,i}^2))\right\|_{V_{s-1}^{-1}}^2\right)\right)=\log\det\left(\mathbf{I}+\frac{\kappa_\mu}{\lambda}\mathbf{K}_t\right)$$

$$=\log\det\left(\mathbf{I}+\frac{\kappa_\mu}{\lambda}\mathbf{J}_t^\top\mathbf{J}_t\right)$$

$$=\log\det\left(\mathbf{I}+\frac{\kappa_\mu}{\lambda}\mathbf{J}_t\mathbf{J}_t^\top\right)$$

$$=\log\det\left(\mathbf{I}+\frac{\kappa_\mu}{\lambda}\sum_{s=1}^{t}\sum_{i=1}^{k}((\phi(x_{s,i}^1)-\phi(x_{s,i}^2))((\phi(x_{s,i}^1)-\phi(x_{s,i}^2))^\top\frac{1}{m}\right) \tag{58}$$

$$\leq\log\det\left(\frac{\kappa_\mu}{\lambda}\mathbf{H}'+\mathbf{I}\right)$$

$$=\widetilde{d}$$

in which we have followed the analysis of Eq. (49) and Eq. (50) in the last inequality.

Combining the results from Eq.(57) and Eq.(58), we have that

$$\sum_{t=1}^{T}\sum_{i=1}^{k}\left\|\frac{1}{\sqrt{m}}(\phi(x_{t,i}^1)-\phi(x_{t,i}^2))\right\|_{V_{t-1}^{-1}}^2\leq 2c_0\sum_{t=1}^{T}\sum_{i=1}^{k}\log\left(1+\frac{\kappa_\mu}{\lambda}\frac{\lambda}{\kappa_\mu}\left\|\frac{1}{\sqrt{m}}(\phi(x_{t,i}^1)-\phi(x_{t,i}^2))\right\|_{V_{t-1}^{-1}}^2\right)$$

$$\leq 2c_0\widetilde{d}$$

Then we can derive an upper bound on the cumulative regret:

*Proof.* Recall that we have

$$reg_t\leq 3(\beta_T+B\sqrt{\frac{\lambda}{\kappa_\nu}}+1)\sum_{i=1}^{k}\left\|\frac{1}{\sqrt{m}}(\phi(\mathbf{x}_{t,i}^1)-\phi(\mathbf{x}_{t,i}^2))\right\|_{V_{t-1}^{-1}}+4k\epsilon_{m,t}' \tag{59}$$

Then we can get an upper bound of the cumulative reward $R_T$

$$Reg_T=\sum_{t=1}^{T}reg_t\leq\sum_{t=1}^{T}3(\beta_T+B\sqrt{\frac{\lambda}{\kappa_\nu}}+1)\sum_{i=1}^{k}\left\|\frac{1}{\sqrt{m}}(\phi(\mathbf{x}_{t,i}^1)-\phi(\mathbf{x}_{t,i}^2))\right\|_{V_{t-1}^{-1}}+4kT\epsilon_{m,t}'$$

$$\leq 3(\beta_T+B\sqrt{\frac{\lambda}{\kappa_\nu}}+1)\sqrt{Tk\sum_{t=1}^{T}\sum_{i=1}^{k}\left\|\frac{1}{\sqrt{m}}(\phi(\mathbf{x}_{t,i}^1)-\phi(\mathbf{x}_{t,i}^2))\right\|_{V_{t-1}^{-1}}^2}+4kT\epsilon_{m,T}'$$

$$\leq 3(\beta_T+B\sqrt{\frac{\lambda}{\kappa_\nu}}+1)\sqrt{Tk2c_0\frac{\lambda}{\kappa_\mu}\widetilde{d}}+4kT\epsilon_{m,T}' \tag{60}$$

Note that $\epsilon_{m,t}'=C_2m^{-1/6}\sqrt{\log m}L^3(\frac{t}{\lambda})^{4/3}$. As long as $m$ satisfy the condition in equation 34, we can get $4kT\epsilon_{m,T}'\leq 1$. And $\beta_T=\widetilde{O}(\frac{1}{\kappa_\mu}\widetilde{d})$, so we have

$$Reg_T\leq 3(\beta_T+B\sqrt{\frac{\lambda}{\kappa_\nu}}+1)\sqrt{Tk2c_0\widetilde{d}}+1=\widetilde{O}\left(\left(\frac{1}{\kappa_\mu}\sqrt{\widetilde{d}}+B\sqrt{\frac{\lambda}{\kappa_\nu}}\right)\sqrt{Tk\widetilde{d}}\right).$$

$\square$

# F  DISCUSSION OF LIPSCHITZ CONTINUITY ASSUMPTION

In this section, we further analyze the reward function under different assumptions. In particular, we focus on the widely used Lipschitz continuity assumption in combinatorial bandit settings and discuss the resulting regret bounds under this assumption.

In the combinatorial dueling bandits setting, our goal is to select the best second super arm:

$$S_t^2 = \arg \max_{S \in \mathcal{S}} [f(S, \mathbf{r}_t) + \frac{\beta_t}{\kappa_\mu} \sigma_{\mathcal{X}_t}(S, S_t^1)]$$

Under the additive reward function assumption, selecting the second super arm becomes efficient, since both the reward function and the confidence interval term can be decomposed into the following components:

$$f(S_t^2, \mathbf{r}_t) + \frac{\beta_t}{\kappa_\mu} \sigma_{\mathcal{X}_t}(S_t^2, S_t^1) = \sum_{i=1}^{k} \left( r_t(\mathbf{x}_{t,i}^2) + \frac{\beta_t}{\kappa_\mu} \left\| \mathbf{x}_{t,i}^1 - \mathbf{x}_{t,i}^2 \right\|_{V_{t-1}^{-1}} \right)$$

$$= \sum_{i=1}^{k} \text{value}(\mathbf{x}_{t,i}^1, \mathbf{x}_{t,i}^2).$$

Each term $\text{value}(\mathbf{x}_{t,i}^1, \mathbf{x}_{t,i}^2)$ depends on both $\mathbf{x}_{t,i}^1$ and $\mathbf{x}_{t,i}^2$, and the total value is obtained by aggregating these terms. Hence, the selection of arms in the second super arm $S_t^2$ can be formulated based on these pairwise values and efficiently solved using the Hungarian Algorithm.

Under the Lipschitz continuity assumption, we aim to select the optimal second super arm such that

$$S_t^2 = \arg \max_{S \in \mathcal{S}} [f(S, \mathbf{r}_t) + \frac{\beta_t}{\kappa_\mu} \sigma_{\mathcal{X}_t}(S, S_t^1)].$$

However, we have no prior knowledge of the reward function $f$, and the term $\sigma_{\mathcal{X}_t}(S, S_t^1)$ under the Lipschitz continuity assumption is also complex. Thus, in order to identify the optimal super arm, we need to evaluate all possible $S \in \mathcal{S}$. Here, we assume access to an efficient oracle that can assist in selecting the second super arm.

## F.1  ASSUMPTIONS

Given the context set $\mathcal{X}_t = \{\mathbf{x}_{t,1}, \mathbf{x}_{t,2}, \ldots, \mathbf{x}_{t,N}\}$, the score vector $\mathbf{r}_t = [r_{t,i}]_{i=1}^N = [r_t(\mathbf{x}_{t,i})]_{i=1}^N$, and a super arm $S_t^1 = \{s_{t,1}^1, s_{t,2}^1, \ldots, s_{t,k}^1\}$ and the context of the super arm $\mathcal{X}_t(S_t^1) = \{\mathbf{x}_{t,1}^1, \mathbf{x}_{t,2}^1, \ldots, \mathbf{x}_{t,k}^1\}$, the common assumptions about $f(S, \mathbf{r})$ are monotonicity and Lipschitz continuity. According to the latter assumption, for any two reward functions $\mathbf{r}$ and $\mathbf{r}'$, we have that for any subset $S$ of arms

$$|f(S_t^1, \mathbf{r}) - f(S_t^1, \mathbf{r}')| \leq C \sqrt{\sum_{i \in S} [r(\mathbf{x}_{t,i}^1) - r'(\mathbf{x}_{t,i}^1)]^2}. \tag{61}$$

But in the combinatorial dueling bandits setting, we need the following assumption instead.

**Assumption 3.** *Without loss of generality, we assume the following:*

- *Lipschitz continuity: For any $S_1$ and $S_2$,*

$$| (f(S_1, \mathbf{r}) - f(S_2, \mathbf{r})) - (f(S_1, \mathbf{r}') - f(S_1, \mathbf{r}')) | \leq C \sqrt{\sum_{i=1}^{k} \left[ (r(\mathbf{x}_{t,i}^1) - r(\mathbf{x}_{t,i}^2)) - (r'(\mathbf{x}_{t,i}^1) - r'(\mathbf{x}_{t,i}^2)) \right]^2}.$$

- *Monotonicity: For any $S_1$ and $S_2$, if $r(\mathbf{x}_{t,i}^1) \geq r(\mathbf{x}_{t,i}^2)$ for all $i$, then $f(S_1, \mathbf{r}) \geq f(S_2, \mathbf{r})$*

---

**Algorithm 5** Linear Combinatorial Dueling Bandits for Lipschitz continuity condition (LinCDB)

---

1: Set $V_0 \triangleq \frac{\lambda}{\kappa_\mu}\mathbf{I}$, $\beta_t \triangleq \sqrt{2\log(1/\delta) + d\log(1 + tkD^2\kappa_\mu/(d\lambda))}$.
2: **for** $t = 1, \ldots, T$ **do**
3:     Find $\theta_t = \arg\min_{\theta'} \mathcal{L}_t(\theta')$ equation 1
4:     Choose the first super arm $S_t^1 = \arg\max_{S \in \mathcal{S}} f(S, \mathbf{r}_t)$
5:     Choose the second arm $S_t^2 = \arg\max_{S \in \mathcal{S}} [f(S, \mathbf{r}_t) + \frac{\beta_t}{\kappa_\mu}\sigma_{\mathcal{X}_t}(S, S_t^1)]$.
6:     Observe the preference feedback: $\{y_{t,i} = \mathbb{1}(x_{t,i}^1 \succ x_{t,i}^2)\}_{i=1,\ldots k}$, and update history
7:     Update $V_t \leftarrow V_{t-1} + \sum_{i=1}^k \widetilde{\mathbf{x}}_{t,i}\widetilde{\mathbf{x}}_{t,i}^\top$
8: **end for**

---

## F.2 Linear Combinatorial Dueling Bandits

**Lemma 16.** *For any two super arm $S_t^1, S_t^2 \in \mathcal{S}$, Define*

$$\sigma_{\mathcal{X}_t}(S_t^1, S_t^2) \triangleq \sqrt{\sum_{i=1}^k \left\|\mathbf{x}_{t,i}^1 - \mathbf{x}_{t,i}^2\right\|_{V_{t-1}^{-1}}^2}.$$

*Then we have that*

$$\left| \left(f(S_t^1, \mathbf{r}^*) - f(S_t^2, \mathbf{r}^*)\right) - \left(f(S_t^1, \mathbf{r}_t) - f(S_t^2, \mathbf{r}_t)\right) \right| \leq \frac{\beta_t}{\kappa_\mu}\sigma_{\mathcal{X}_t}(S_t^1, S_t^2).$$

*Proof.*

$$\left| \left(f(S_t^1, \mathbf{r}^*) - f(S_t^2, \mathbf{r}^*)\right) - \left(f(S_t^1, \mathbf{r}_t) - f(S_t^2, \mathbf{r}_t)\right) \right|$$

$$\overset{(a)}{\leq} C\sqrt{\sum_{i=1}^k \left[\left(r^*(\mathbf{x}_{t,i}^1) - r^*(\mathbf{x}_{t,i}^2)\right) - \left(r_t(\mathbf{x}_{t,i}^1) - r_t(\mathbf{x}_{t,i}^2)\right)\right]^2}.$$

$$\overset{(b)}{\leq} C\frac{\beta_t}{\kappa_\mu}\sqrt{\sum_{i=1}^k \left\|\mathbf{x}_{t,i}^1 - \mathbf{x}_{t,i}^2\right\|_{V_{t-1}^{-1}}^2} \tag{62}$$

$$= C\frac{\beta_t}{\kappa_\mu}\sigma_{\mathcal{X}_t}(S_t^1, S_t^2).$$

where (a) follows from Assumption 3, (b) follows from Lemma 1.

$\square$

**Lemma 17.** $\sigma_{\mathcal{X}_t}(S_t^*, S_t^2) \leq \sigma_{\mathcal{X}_t}(S_t^*, S_t^1) + \sigma_{\mathcal{X}_t}(S_t^1, S_t^2)$

*Proof.* To begin with, we have that

$$\left\|\mathbf{x}_{t,i}^* - \mathbf{x}_{t,i}^2\right\|_{V_{t-1}^{-1}} \leq \left\|\mathbf{x}_{t,i}^* - \mathbf{x}_{t,i}^1\right\|_{V_{t-1}^{-1}} + \left\|\mathbf{x}_{t,i}^1 - \mathbf{x}_{t,i}^2\right\|_{V_{t-1}^{-1}}$$

This leads to

$$\left\|\mathbf{x}_{t,i}^* - \mathbf{x}_{t,i}^2\right\|_{V_{t-1}^{-1}}^2 \leq \left\|\mathbf{x}_{t,i}^* - \mathbf{x}_{t,i}^1\right\|_{V_{t-1}^{-1}}^2 + \left\|\mathbf{x}_{t,i}^1 - \mathbf{x}_{t,i}^2\right\|_{V_{t-1}^{-1}}^2 + 2\left\|\mathbf{x}_{t,i}^* - \mathbf{x}_{t,i}^1\right\|_{V_{t-1}^{-1}} \left\|\mathbf{x}_{t,i}^1 - \mathbf{x}_{t,i}^2\right\|_{V_{t-1}^{-1}}$$

$$\sum_{i=1}^k \left\|\mathbf{x}_{t,i}^* - \mathbf{x}_{t,i}^2\right\|_{V_{t-1}^{-1}}^2 \leq \sum_{i=1}^k \left\|\mathbf{x}_{t,i}^* - \mathbf{x}_{t,i}^1\right\|_{V_{t-1}^{-1}}^2 + \sum_{i=1}^k \left\|\mathbf{x}_{t,i}^1 - \mathbf{x}_{t,i}^2\right\|_{V_{t-1}^{-1}}^2 + 2\sum_{i=1}^k \left\|\mathbf{x}_{t,i}^* - \mathbf{x}_{t,i}^1\right\|_{V_{t-1}^{-1}} \left\|\mathbf{x}_{t,i}^1 - \mathbf{x}_{t,i}^2\right\|_{V_{t-1}^{-1}}$$

$$\leq \sum_{i=1}^k \left\|\mathbf{x}_{t,i}^* - \mathbf{x}_{t,i}^1\right\|_{V_{t-1}^{-1}}^2 + \sum_{i=1}^k \left\|\mathbf{x}_{t,i}^1 - \mathbf{x}_{t,i}^2\right\|_{V_{t-1}^{-1}}^2 + 2\sqrt{\sum_{i=1}^k \left\|\mathbf{x}_{t,i}^* - \mathbf{x}_{t,i}^1\right\|_{V_{t-1}^{-1}}^2}\sqrt{\sum_{i=1}^k \left\|\mathbf{x}_{t,i}^1 - \mathbf{x}_{t,i}^2\right\|_{V_{t-1}^{-1}}^2}$$

$$= \left(\sqrt{\sum_{i=1}^k \left\|\mathbf{x}_{t,i}^* - \mathbf{x}_{t,i}^1\right\|_{V_{t-1}^{-1}}^2} + \sqrt{\sum_{i=1}^k \left\|\mathbf{x}_{t,i}^1 - \mathbf{x}_{t,i}^2\right\|_{V_{t-1}^{-1}}^2}\right)^2$$

$$(63)$$

in which we have applied the Cauchy–Schwarz inequality in the last inequality. Therefore, we have that

$$\sqrt{\sum_{i=1}^k \left\|\mathbf{x}_{t,i}^* - \mathbf{x}_{t,i}^2\right\|_{V_{t-1}^{-1}}^2} \leq \sqrt{\sum_{i=1}^k \left\|\mathbf{x}_{t,i}^* - \mathbf{x}_{t,i}^1\right\|_{V_{t-1}^{-1}}^2} + \sqrt{\sum_{i=1}^k \left\|\mathbf{x}_{t,i}^1 - \mathbf{x}_{t,i}^2\right\|_{V_{t-1}^{-1}}^2} \qquad (64)$$

That is,

$$\sigma_{\mathcal{X}_t}(S_t^*, S_t^2) \leq \sigma_{\mathcal{X}_t}(S_t^*, S_t^1) + \sigma_{\mathcal{X}_t}(S_t^1, S_t^2). \qquad (65)$$

$\square$

Following the proof of Lemma 5, we can easily establish the following lemma.

**Lemma 18.** *In any iteration $t$, the regret is bound by*

$$reg_t \leq 3C\frac{\beta_t}{\kappa_\mu}\sigma_{\mathcal{X}_t}(S_t^1, S_t^2) = 3C\frac{\beta_t}{\kappa_\mu}\sqrt{\sum_{i=1}^k \left\|\mathbf{x}_{t,i}^1 - \mathbf{x}_{t,i}^2\right\|_{V_{t-1}^{-1}}^2}$$

And then we can get the total regret $Reg_T$.

**Theorem 3.** *Let $\beta_t \triangleq \sqrt{2\log(1/\delta) + d\log\left(1 + tkD^2\kappa_\mu/(d\lambda)\right)}$ and $\lambda \leq \frac{\kappa_\mu}{D^2}$, then With probability of at least $1 - \delta$, we have that*

$$Reg_T \leq \frac{3}{\kappa_\mu}\sqrt{2\log(1/\delta) + d\log\left(1 + TkD^2\kappa_\mu/(d\lambda)\right)}\sqrt{2Tkd\log(1 + TkD^2\kappa_\mu/(d\lambda))}$$

*Proof.*

$$Reg_T = \sum_{t=1}^T reg_t \leq \sum_{t=1}^T 3C\frac{\beta_t}{\kappa_\mu}\sqrt{\sum_{i=1}^k \left\|\mathbf{x}_{t,i}^1 - \mathbf{x}_{t,i}^2\right\|_{V_{t-1}^{-1}}^2} \qquad (66)$$

$$\leq 3C\frac{\beta_t}{\kappa_\mu}\sqrt{T\sum_{t=1}^T\sum_{i=1}^k \left\|\mathbf{x}_{t,i}^1 - \mathbf{x}_{t,i}^2\right\|_{V_{t-1}^{-1}}^2} \qquad (67)$$

$$\leq 3C\frac{\beta_t}{\kappa_\mu}\sqrt{2Td\log(1 + \frac{\kappa_\mu TkD^2}{d\lambda})} \qquad (68)$$

$$= 3C\frac{\beta_t}{\kappa_\mu}\sqrt{2Td\log(1 + \frac{\kappa_\mu TkD^2}{d\lambda})} \qquad (69)$$

So we have that $Reg_T \leq \frac{3C}{\kappa_\mu}\left[\sqrt{4Td\log(1/\delta)\log(1 + \kappa_\mu TkD^2/(d\lambda))} + \sqrt{2T}d\log(1 + \frac{\kappa_\mu TkL^2}{d\lambda})\right]$

Ignoring all log factors, we have that $Reg_T = \widetilde{O}(\frac{1}{\kappa_\mu}d\sqrt{T})$

$\square$

F.3  NEURAL COMBINATORIAL DUELING BANDITS

---

**Algorithm 6** Neural Combinatorial Dueling Bandits for Lipschitz continuity condition (NCDB)

---

1: Set $V_0 \triangleq \frac{\lambda}{\kappa_\mu}\mathbf{I}$, $\beta_T \triangleq \frac{1}{\kappa_\mu}\sqrt{\widetilde{d} + 2\log(1/\delta)}$ ($\widetilde{d}$ is defined in Definition 1), $\nu_T \triangleq \left(\beta_T + B\sqrt{\frac{\lambda}{\kappa_\mu}} + 1\right)\frac{\kappa_\mu}{\lambda}$.
2: **for** $t = 1, \ldots, T$ **do**
3:     Train NN using history $\{(\mathbf{x}_{s,i}^1, \mathbf{x}_{s,i}^2, y_{s,i})\}_{s=1,i=1}^{t-1,k}$ by minimizing loss function equation 6
4:     Receive the contexts $\mathcal{X}_t$
5:     Compute $r_t(x) = h(x; \theta_t)$
6:     Choose the first arm set $S_t^1 = \arg\max_{S \in \mathcal{S}} f(S, \mathbf{r}_t)$
7:     Choose the second arm set $S_t^2 = \arg\max_{S \in \mathcal{S}} f(S, \mathbf{r}_t) + \nu_T \sigma_{t-1}(S, S_t^1)$
8:     Observe the preference feedback: $\{y_{t,i} = \mathbb{K}(x_{t,i}^1 \succ x_{t,i}^2)\}_{i=1,\ldots k}$, and update history
9: **end for**

---

**Definition 4.** *Define* $\sigma_t(\mathbf{x}, \mathbf{x}') \triangleq \left\|\frac{1}{\sqrt{m}}(\phi(\mathbf{x}) - \phi(\mathbf{x}'))\right\|_{V_{t-1}^{-1}} \left(\beta_T + B\sqrt{\frac{\lambda}{\kappa_\mu}} + 1\right) + 2\epsilon'_{m,t}$ *and for two sets super arm* $S_t^1 = \{s_{t,1}^1, s_{t,2}^1, \ldots, s_{t,k}^1\}$, *the context* $\mathcal{X}_t(S_t^1) = \{\mathbf{x}_{t,1}^1, \mathbf{x}_{t,2}^1, \ldots, \mathbf{x}_{t,k}^1\}$ *and super arm* $S_t^2 = \{s_{t,1}^2, s_{t,2}^2, \ldots, s_{t,k}^2\}$ *and the context* $\mathcal{X}_t(S_t^2) = \{\mathbf{x}_{t,1}^2, \mathbf{x}_{t,2}^2, \ldots, \mathbf{x}_{t,k}^2\}$ *define*
$\sigma_t(S_1, S_2) \triangleq \sqrt{\sum_{i=1}^k \sigma_t^2(\mathbf{x}_{t,i}^1, \mathbf{x}_{t,i}^2)}$

**Lemma 19.** *For any super arm* $S_t^1, S_t^2 \in \mathcal{S}$.

$$\left|\left(f(S_t^1, \mathbf{r}^*) - f(S_t^2, \mathbf{r}^*)\right) - \left(f(S_t^1, \mathbf{r}_t) - f(S_t^2, \mathbf{r}_t)\right)\right| \le C\sigma_t(S_1, S_2)$$

*Proof.*

$$\left|\left(f(S_t^1, \mathbf{r}^*) - f(S_t^2, \mathbf{r}^*)\right) - \left(f(S_t^1, \mathbf{r}_t) - f(S_t^2, \mathbf{r}_t)\right)\right|$$

$$\overset{(a)}{\le} C\sqrt{\sum_{i=1}^k \left[\left(r^*(\mathbf{x}_{t,i}^1) - r^*(\mathbf{x}_{t,i}^2)\right) - \left(r_t(\mathbf{x}_{t,i}^1) - r_t(\mathbf{x}_{t,i}^2)\right)\right]^2}$$

$$\overset{(b)}{=} C\sqrt{\sum_{i=1}^k \left[\left(r^*(\mathbf{x}_{t,i}^1) - r^*(\mathbf{x}_{t,i}^2)\right) - [h(\mathbf{x}_{t,i}^1; \theta_t) - h(\mathbf{x}_{t,i}^2; \theta_t)]\right]^2} \qquad (70)$$

$$\overset{(c)}{\le} C\sqrt{\sum_{i=1}^k \left(\left\|\frac{1}{\sqrt{m}}(\phi(\mathbf{x}_{t,i}^1) - \phi(\mathbf{x}_{t,i}^2))\right\|_{V_{t-1}^{-1}} (\beta_T + B\sqrt{\frac{\lambda}{\kappa_\mu}} + 1) + 2\epsilon'_{m,t}\right)^2}$$

$$\overset{(d)}{=} C\sigma_t(S_1, S_2).$$

where step (a) follows from Assumption 3, step (b) follows from the definition of reward function $r_t$, step (c) follows from Lemma 15 and step (d) follows from Definition 3.

$\square$

**Lemma 20.** *In any iteration* $t$, *the regret is bounded by*

$$reg_t \le 6C(\beta_T + B\sqrt{\frac{\lambda}{\kappa_\mu}} + 1)\sqrt{\sum_{i \in [k]} \left\|\frac{1}{\sqrt{m}}(\phi(\mathbf{x}_{t,i}^1) - \phi(\mathbf{x}_{t,i}^2))\right\|_{V_{t-1}^{-1}}^2} + 12C\sqrt{k}\epsilon'_{m,t}$$

To begin with, we have

$$
\begin{aligned}
reg_t &= 2\mathrm{opt}_{\mathbf{r}^*} - \big(f(S_t^1, \mathbf{r}^*) + f(S_t^2, \mathbf{r}^*)\big) \\
&= f(S_t^*, \mathbf{r}^*) - f(S_t^1, \mathbf{r}^*) + f(S_t^*, \mathbf{r}^*) - f(S_t^2, \mathbf{r}^*) \\
&\overset{(a)}{\leq} f(S_t^*, \mathbf{r}_t) - f(S_t^1, \mathbf{r}_t) + C\sigma_t(S^*, S_1) + f(S_t^*, \mathbf{r}_t) - f(S_t^2, \mathbf{r}_t) + C\sigma_t(S^*, S_2) \\
&\overset{(b)}{\leq} f(S_t^*, \mathbf{r}_t) - f(S_t^1, \mathbf{r}_t) + C\sigma_t(S^*, S_1) + \\
&\quad\ f(S_t^*, \mathbf{r}_t) - f(S_t^1, \mathbf{r}_t) + f(S_t^1, \mathbf{r}_t) - f(S_t^2, \mathbf{r}_t) + C\sigma_t(S^*, S_1) + C\sigma_t(S_1, S_2) \\
&= 2\big(f(S_t^*, \mathbf{r}_t) - f(S_t^1, \mathbf{r}_t) + C\sigma_t(S^*, S_1)\big) + f(S_t^1, \mathbf{r}_t) - f(S_t^2, \mathbf{r}_t) + C\sigma_t(S_1, S_2) \\
&\overset{(c)}{\leq} 2\big(f(S_t^2, \mathbf{r}_t) - f(S_t^1, \mathbf{r}_t) + C\sigma_t(S_2, S_1)\big) + f(S_t^1, \mathbf{r}_t) - f(S_t^2, \mathbf{r}_t) + C\sigma_t(S_1, S_2) \\
&= f(S_t^2, \mathbf{r}_t) - f(S_t^1, \mathbf{r}_t) + 3C\sigma_t(S_2, S_1) \\
&\overset{(d)}{\leq} 3C\sigma_t(S_1, S_2) \\
&= 3C\sqrt{\sum_{i=1}^{k}\left(\left\|\frac{1}{\sqrt{m}}(\phi(\mathbf{x}_{t,i}^1) - \phi(\mathbf{x}_{t,i}^2))\right\|_{V_{t-1}^{-1}}\left(\beta_T + B\sqrt{\frac{\lambda}{\kappa_\mu}} + 1\right) + 2\epsilon'_{m,t}\right)^2} \\
&\overset{(e)}{\leq} 3C\sqrt{\sum_{i=1}^{k}\left(2\max\left\{\left\|\frac{1}{\sqrt{m}}(\phi(\mathbf{x}_{t,i}^1)) - \phi(\mathbf{x}_{t,i}^2)))\right\|_{V_{t-1}^{-1}}\left(\beta_T + B\sqrt{\frac{\lambda}{\kappa_\mu}} + 1\right), 2\epsilon'_{m,t}\right\}\right)^2}
\end{aligned}
\tag{71}
$$

Step (a) follows from Eq. (53), step (b) follows from the fact that $\sigma_{t-1}(S_t^*, S_t^2) \leq \sigma_{t-1}(S_t^*, S_t^1) + \sigma_{t-1}(S_t^1, S_t^2)$ using triangle inequality, step (c) follows from the way in which $S_2$ is selected: $S_{t,2} = \arg\max_{S \in \mathcal{S}} f_{\mathbf{r}_t, \mathcal{X}_t}(S) + C\sigma_t(S, S_{t,1})$, step (d) follows from the way in which $S_1$ is selected: $S_t^1 = \arg\max_{S \in \mathcal{S}} f_{\mathbf{r}_t, \mathcal{X}_t}(S)$. and step (e) follows from the fact that $a + b \leq 2\max\{a, b\}$

If we denote $\mathcal{B}_i = \left\|\frac{1}{\sqrt{m}}(\phi(\mathbf{x}_{t,i}^1) - \phi(\mathbf{x}_{t,i}^2))\right\|_{V_{t-1}^{-1}}\left(\beta_T + B\sqrt{\frac{\lambda}{\kappa_\mu}} + 1\right)$, then we have

$$
\begin{aligned}
&\sqrt{\sum_{i=1}^{k}\left(2\max\left\{\left\|\frac{1}{\sqrt{m}}(\phi(\mathbf{x}_{t,i}^1) - \phi(\mathbf{x}_{t,i}^2))\right\|_{V_{t-1}^{-1}}\left(\beta_T + B\sqrt{\frac{\lambda}{\kappa_\mu}} + 1\right), 2\epsilon'_{m,t}\right\}\right)^2} = \sqrt{4\left(\sum_{\mathcal{B}_i \geq 2\epsilon'_{m,t}}\mathcal{B}_i^2 + \sum_{\mathcal{B}_i < 2\epsilon'_{m,t}} 4\epsilon'^2_{m,t}\right)} \\
&\leq 2\sqrt{\sum_{i \in [k]}\mathcal{B}_i^2 + \sum_{i \in [k]} 4\epsilon'^2_{m,t}} \\
&\leq 2\sqrt{\sum_{i \in [k]}\mathcal{B}_i^2} + 2\sqrt{\sum_{i \in [k]} 4\epsilon'^2_{m,t}} \\
&= 2\sqrt{\sum_{i \in [k]}\mathcal{B}_i^2} + 4\sqrt{k}\epsilon'_{m,t}
\end{aligned}
\tag{72}
$$

By substituting Eq.72, we have

$$
reg_t \leq 6C \left( \sqrt{ \sum_{i \in [k]} \left( \left\| \frac{1}{\sqrt{m}}(\phi(\mathbf{x}_{t,i}^1) - \phi(\mathbf{x}_{t,i}^2)) \right\|_{V_{t-1}^{-1}} (\beta_T + B\sqrt{\frac{\lambda}{\kappa_\mu}} + 1) \right)^2 } + 2\sqrt{k}\epsilon_{m,t}' \right)
$$

$$
= 6C \left( (\beta_T + B\sqrt{\frac{\lambda}{\kappa_\mu}} + 1) \sqrt{ \sum_{i \in [k]} \left\| \frac{1}{\sqrt{m}}(\phi(\mathbf{x}_{t,i}^1) - \phi(\mathbf{x}_{t,i}^2)) \right\|_{V_{t-1}^{-1}}^2 } + 2\sqrt{k}\epsilon_{m,t}' \right) \tag{73}
$$

$$
= 6C(\beta_T + B\sqrt{\frac{\lambda}{\kappa_\mu}} + 1) \sqrt{ \sum_{i \in [k]} \left\| \frac{1}{\sqrt{m}}(\phi(\mathbf{x}_{t,i}^1) - \phi(\mathbf{x}_{t,i}^2)) \right\|_{V_{t-1}^{-1}}^2 } + 12C\sqrt{k}\epsilon_{m,t}'
$$

Then we can derive an upper bound on the cumulative regret:

**Theorem 4.** *Let $\beta_t \triangleq \sqrt{2\log(1/\delta) + d\log(1 + tkD^2\kappa_\mu/(d\lambda))}$ and $\lambda \leq \frac{\kappa_\mu}{D^2}$, then With probability of at least $1 - \delta$, we have that*

$$
Reg_T \leq 6C(\sqrt{2\log(1/\delta) + d\log(1 + tkD^2\kappa_\mu/(d\lambda))} + B\sqrt{\frac{\lambda}{\kappa_\mu}} + 1)\sqrt{T2c_0\frac{\lambda}{\kappa_\mu}\widetilde{d}} + 12CT\sqrt{k}\epsilon_{m,t}'
$$

*Proof.* We can get an upper bound of the cumulative reward $Reg_T$

$$
Reg_T = \sum_{t=1}^T reg_t \leq \sum_{t=1}^T 6C(\beta_T + B\sqrt{\frac{\lambda}{\kappa_\mu}} + 1)\sqrt{ \sum_{i \in [k]} \left\| \frac{1}{\sqrt{m}}(\phi(\mathbf{x}_{t,i}^1) - \phi(\mathbf{x}_{t,i}^2)) \right\|_{V_{t-1}^{-1}}^2 } \tag{74}
$$

$$
+ \sum_{t=1}^T 12C\sqrt{k}\epsilon_{m,t}' \tag{75}
$$

$$
\leq 6C(\beta_T + B\sqrt{\frac{\lambda}{\kappa_\mu}} + 1)\sqrt{ T\sum_{t=1}^T \sum_{i \in [k]} \left\| \frac{1}{\sqrt{m}}(\phi(\mathbf{x}_{t,i}^1) - \phi(\mathbf{x}_{t,i}^2)) \right\|_{V_{t-1}^{-1}}^2 } \tag{76}
$$

$$
+ 12CT\sqrt{k}\epsilon_{m,t}' \tag{77}
$$

$$
\leq 6C(\beta_T + B\sqrt{\frac{\lambda}{\kappa_\mu}} + 1)\sqrt{ T2c_0\frac{\lambda}{\kappa_\mu}\widetilde{d} } + 12CT\sqrt{k}\epsilon_{m,t}' \tag{78}
$$

$$
\tag{79}
$$

Ignoring all log factors, we can get

$$
Reg_T \leq \widetilde{O}\left( \left( \frac{1}{\kappa_\mu}\sqrt{\widetilde{d}} + B\sqrt{\frac{\lambda}{\kappa_\mu}} \right)\sqrt{T\widetilde{d}} \right).
$$

$\square$

