# OpenReview forum: "Combinatorial Dueling Bandits"
_ICLR.cc/2026/Conference — ICLR 2026 Conference Withdrawn Submission_

### Official Review · Reviewer_DbVL · 2025-10-27

**Soundness:** 3
**Presentation:** 2
**Contribution:** 2
**Rating:** 4
**Confidence:** 4

**Summary:**

The paper introduces the combinatorial dueling bandit, studying the preference feedback. It proposes algorithms in both linear and neural settings and provides theoretical guarantees.

**Strengths:**

(1) The paper is well motivated by real-life applications like recommendation systems and LLMs. The theoretical results are solid, with a well-written introduction to the main contents.

(2) It circumvents the computational hurdle in the algorithm by framing the selection of the second super arm as a bipartite matching problem and applying the Hungarian algorithm.

(3) It is the first work to study combinatorial bandits with preference feedback.

**Weaknesses:**

(1) The selection of the second super arm with the Hungarian algorithm is ambiguous. In the introduction, this part is claimed as one of the key technical challenges of this paper. However, this part is not emphasized enough in the main text, even the Hungarian algorithm is not introduced. Even after reading the appendix, some key concepts are not well established, like the “bipartite matching task”. A brief introduction of what is studied is necessary for the completeness of the paper.

Moreover, the computational complexity of the algorithm has not been discussed, which makes it unclear for readers to understand the improvement of this algorithm design.

(2) The analysis techniques are not novel, especially for the bandit part. The bipartite matching task, v.s. the super arm selection, is good, yet still well established in the graph theory field, I believe. The bandit part is almost the same as previous works, for example, [1] for linear bandits, [2] for neural bandits. The similarity includes both algorithm design and regret analysis. What’s the technical novelty, except for replacing all arms with super arms?

(3) The feedback studied in this paper is what is usually called semi-bandit feedback. However, the key term semi-bandit feedback is not emphasized in this paper. In addition, the setting studies do not consider the special property of dueling bandits. For example, what will happen when we only know how many arms win in the first super arm, instead of which arms (bandit feedback)? Or what if we can know all the comparisons between two super arms, (an N $\times$ N preference matrix). Will this additional information bring benefit to the final result? Without these kinds of information for dueling bandit, it seems a direct generalization from combinatorial bandit, instead of a complete analysis of combinatorial dueling bandits.

(4) Is the notation of the indicator function in Algorithm 1, Line 1, wrongly presented?

[1] Stochastic contextual dueling bandits under linear stochastic transitivity models, ICML

[2] Neural dueling bandits: Preference-based optimization with human feedback, ICLR

**Questions:**

(1) In [3], the $\kappa$ dependence can be removed (or moved to the low-order terms). Do you think it’s also possible in the combinatorial setting?

(2) What is the lower bound for linear combinatorial dueling bandit? Is it the same as the combinatorial bandit and dueling bandit separately?

[3] Nearly optimal algorithms for contextual dueling bandits from adversarial feedback, ICML

---

### Official Review · Reviewer_gNsm · 2025-10-31

**Soundness:** 3
**Presentation:** 3
**Contribution:** 2
**Rating:** 2
**Confidence:** 4

**Summary:**

This work initiates a study of contextual combinatorial dueling bandits where the learner selects two sets of arms and observes pairwise comparison feedback between arms in the two sets. They prove new regret upper bounds under two score models (linear and neural network) on preferences.

**Strengths:**

* This appears to be the first work to study a combinatorial variant of the dueling bandit problem and, as such, demonstrates the first regret upper bounds in this setting.
* The problem is well-motivated and the presentation is straightforward to follow.
* There are many synthetic experiments to validate theoretical results.

**Weaknesses:**

My main concern is with the scope of technical novelty for this problem, and how the modeling assumptions seem to make the analysis a straightforward extension of known techniques.

First, the BTL model, while well-studied, is restrictive and gives a parametric form on score functions which allows straightforward estimation of per-arm scores. It would be more interesting if the learner had to estimate the general preference matrix (as done in more general works on dueling bandits) under standard assumptions such as SST and/or STI under combinatorial feedback. The preference feedback does not strongly come into play under such a model and, for ordinary dueling bandits, there is a very expansive literature on generalizing these assumptions (see Bengs et al., 2018, "Preference-based Online Learning with Dueling Bandits: A Survey" for details).

Second, the nature of the problem setting is that the learner selects two ordered equal-size subsets of arms (dubbed "super-arms") and observes ordered preference feedback of comparisons between arms. Furthermore, the feeedback is semi-bandit and the reward function model devolves into a sum of scores among chosen arms. For me, this, especially when combined with the BTL model, already makes the problem very similar to regular BTL-model dueling bandits. The combinatorial feedback just seems to enforce a higher batch-size on selecting arms. It would be more challenging and interesting to study a subset-based feedback where the learner observes a bandit-feedback preference of set A to set B, properly defined in some well-motivated way (e.g., if all arms in A beat all arms in B). In my opinion, this would be closer to the spirit of combinatorial bandits. In my view, these utitlity-based modeling assumptions as well as the linear model are what make the "efficient selection of the second super-arm" (which the authors purport is the main technical challenge to overcome) possible.

It is also unclear the tightness of regret bounds in this setting, and the proper dependence on dimension, super-arm size, and other parameters.

**Questions:**

Please see weaknesses above.

---

### Official Review · Reviewer_xbK2 · 2025-11-10

**Soundness:** 3
**Presentation:** 3
**Contribution:** 3
**Rating:** 4
**Confidence:** 4

**Summary:**

The paper introduces a new relative feedback model for (contextual) dueling bandits with subsets as decisions, and goes on to develop subset selection algorithms with regret guarantees. In this model, $N$ linear contextual arms become available to the agent at each instant. The agent must select 2 size-$k$ ordered subsets from among them. Then, each item from the first subset duels with the corresponding item from the second subset, according to a standard BTL model, and $k$ dueling feedback bits are generated as feedback from the agent. The agent's performance is measured as (dueling) regret with the subset-wise reward being an aggregate of its constituent items' rewards. In a way, this model is the relative-feedback counterpart of the combinatorial semi-bandit in the absolute-reward setting, where a single $k$-subset is chosen and all $k$ absolute rewards are individually observed by the agent. A subset selection algorithm (LinCDB) is proposed which involves selecting the first subset 'greedily' as per the MLE of the linear-BTL model, and the second subset as the "most optimistic competitor" to the first subset -- this is shown to be solvable in time polynomial in the arm set via weighted matching / max-flow, with a Hungarian algorithm implementation. The paper then generalizes the linear BTL score function to nonlinear neural functions and proceeds along similar lines.

**Strengths:**

The primary strength of the paper is that it introduces a new feedback model for relative utility-based adaptive decision-making, where k pairwise comparisons can be requested. This generalizes standard (by now) contextual dueling bandits which are obtained with $k=1$.

In terms of the quality and significance of the contribution, the paper contains a substantial amount of technical results, covering the spectrum from arguably "simple" linear feature models to neural network-based nonlinear models (albeit in the stylized "kernelized" sense via the neural tangent kernel regime). The fact that the paper manages to find an efficient (in arm space) implementation for their algorithm via a connection to max-weighted matching is quite impressive and valuable from a practical standpoint.

The paper's quality of technical exposition is high, with all necessary technical preliminaries introduced cleanly. It also presents the main algorithms and their performance guarantees, along with their derivations, in a transparent manner.

**Weaknesses:**

The new decision-and-feedback model of the paper (k duels via 2 size-k ordered subsets of N items), while being introduced for the first time in the dueling bandit literature, lacks sufficient motivation or real-world context. Where, for instance, in a real-world application, could this kind of subset building followed by win-loss feedback design be relevant? Are there documented settings, e.g., in the recommender systems community or its literature, where such feedback is realistic? (for instance, many generative AI chatbots these days ask for relative preferences over a pair of responses to a prompt. How feasible or burdensome might it be if k-pairs are needed to be dueled via a single human user?) I am fine if the setting is modeling a system that is futuristic, but I feel that adequate discussion should be present about a path from current technology to such data collection pipelines.

On a related note, assuming that the system that provides k-pair relative feedback is a different module than the actual decision making agent (e.g., a crowdsourcing system that contacts individual workers for their opinions on pairs), why restrict oneself in the model to providing 2 k-length subsets out of which pairs are made by going down the order in each subset? Why not envision a more general decision model where the agent can simply propose k feedback pairs to receive feedback for, in a more flexible model, without building 2 subsets and then generating pairs based on them?

The final aspect about the modeling that lacks clarity is the specific choice of performance metric (dueling regret and its variants). Is there a concrete interpretation of the average / weak instantaneous regret, together with the additive structure within the subset, in an application sense? Does it corresponding to an actual loss of utility as desired in an application? More discussion or clarity on this would be extremely helpful to add.

Related work and approaches: While I appreciated that the paper adequately discusses relevant work in the appendix, there is not adequate discussion about how the proposed algorithm based on the "select first subset greedily according to the MLE, select second subset as the most optimistic competitor" idea differs / generalises / resembles other decision-making principles in prior work. For instance, at least when $k=1$, how does the authors' design of a pair to be queried compare with approaches like MaxInP (Saha 2021) and COLSTIM (Bengs et al, 2022)? From what I see, the idea of a "toughest" or "most optimistic" competing subset is already present in these approaches. Broadly, I would like to see more comparisons or discussion with competing approaches in the $k=1$ setting in order for the paper's decision principle to stand out in a scientific sense.

Numerical experiments: Again, if there are other algorithmic approaches that apply for the $k=1$ setting of this paper (standard contextual dueling bandits), e.g., the ones mentioned above, then can't any of them be 'naturally' extended to the $k>1$ setting of this paper via the following approach?: To select $k$ pairs, first select one pair via the existing principle. Then, remove this pair from consideration and select another pair from the remaining $N-2$ items by applying the principle again. Repeat. If this is possible, then the author(s) ought to benchmark such 'natural' or 'obvious' extensions as meaningful baselines in addition to only their strategy, in the experiments section.

Several assumptions made in order to derive regret bounds are explained as 'standard in the literature', but I am not able to clearly see their relevance independently, and more elaboration would be useful around them. For instance,
1) Assumption 1 assumes that $k_\mu > 0$. What does this specifically imply when the link function $\mu$ equals the sigmoid as in the case of the BTL model? Is it ruling out the existence of 'equivalent' arms?
2) In Assumption 2, part 3, what is the underlying reason for making the symmetry assumption on the features $x_j$? It seems odd without sufficient context. The paper mentions the existence of an 'easy feature space transformation' to ensure this assumption, but I was unable to find it explicitly mentioned in existing literature. Can the author(s) make this explicit?

**Questions:**

Please see above - questions are embedded naturally within points in the "Weaknesses" section.

---

### Note · Authors · 2026-01-18

I have read and agree with the venue's withdrawal policy on behalf of myself and my co-authors.